# DJ-1 protects proteins from acylation by catalyzing the hydrolysis of highly reactive cyclic 3-phosphoglyceric anhydride

Aizhan Akhmadi [1], Adilkhan Yeskendir [2,3], Nelly Dey[3], Arman Mussakhmetov [4,5], Zariat Shatkenova [6], Arman Kulyyassov [4], Anna Andreeva [6] & Darkhan Utepbergenov [2] ✉

Mutations in the human *PARK7* gene that encodes protein DJ-1 lead to familial Parkinsonism due to loss of dopaminergic neurons. However, the molecular function of DJ-1 underpinning its cytoprotective effects are unclear. Recently, DJ-1 has been shown to prevent acylation of amino groups of proteins and metabolites by 1,3-bisphosphoglycerate. This acylation is indirect and thought to proceed via the formation of an unstable intermediate, presumably a cyclic 3-phosphoglyceric anhydride (cPGA). Several lines of evidence indicate that DJ-1 destroys cPGA, however this enzymatic activity has not been directly demonstrated. Here, we report simple and effective procedures for synthesis and quantitation of cPGA and present a comprehensive characterization of this highly reactive acylating electrophile. We demonstrate that DJ-1 is an efficient cPGA hydrolase with $k_{cat}/K_m = 5.9 \times 10^6$ M$^{-1}$s$^{-1}$. Experiments with DJ-1-null cells reveal that DJ-1 protects against accumulation of 3-phosphoglyceroyl-lysine residues in proteins. Our results establish a definitive cytoprotective function for DJ-1 that uses catalytic hydrolysis of cPGA to mitigate the damage from this glycolytic byproduct.

Mutations in the human gene *PARK7* cause an early onset of Parkinson's disease (PD) due to excessive loss of dopaminergic neurons[1]. *PARK7* encodes a small (~20 kDa) protein called DJ-1 that is present in nearly all organisms from bacteria to mammals. DJ-1 belongs to a diverse DJ-1/ThiJ/PfpI superfamily of proteins some of which possess glyoxalase or hydrolase activities while others remain uncharacterized[2]. The molecular mechanisms underlying the protective effects of DJ-1 in dopaminergic neurons remain unclear. Earlier reports suggested that DJ-1 senses oxidative stress through its readily oxidizable catalytic cysteine and protects cells by orchestrating the cellular defense against reactive oxygen species[3,4]. However, no signaling pathways or binding partners have been unequivocally established for DJ-1, leaving the mechanism of neuroprotection by DJ-1 an open question.

More recently, various enzymatic activities have been ascribed to DJ-1, though none of these adequately explain its neuroprotective function. For example, DJ-1 can inactivate the toxic secondary metabolite methylglyoxal by converting it to a less reactive lactic acid[5–7]. Since methylglyoxal reacts with nucleophilic amino acids to cause a gradual accumulation of advanced glycation end products implicated in various pathologies[8], the methylglyoxalase activity of DJ-1 suggests a mechanistically sound explanation of its neuroprotective effects. However, catalytic constants for methylglyoxalase activity of DJ-1 have been reported in the range of 0.02–0.3 s$^{-1}$[15–7,9], too low to allow

[1]Ph.D. Program in Life Sciences, School of Sciences and Humanities, Nazarbayev University, Astana 010000, Kazakhstan. [2]Department of Chemistry, School of Sciences and Humanities, Nazarbayev University, Astana 010000, Kazakhstan. [3]Master Program, School of Medicine, Nazarbayev University, Astana 010000, Kazakhstan. [4]National Center for Biotechnology, Astana 010000, Kazakhstan. [5]Ph.D. Program in Biology, L.N. Gumilyov Eurasian National University, Astana 010000, Kazakhstan. [6]Department of Biology, School of Sciences and Humanities, Nazarbayev University, Astana 010000, Kazakhstan. ✉e-mail: darkhan.utepbergenov@nu.edu.kz

meaningful physiological function given the ubiquitous presence of vastly more efficient glyoxalase I/II enzymes[10]. Accordingly, many studies have found little or no effect of DJ-1 on protection of cells against methylglyoxal[6,9] or accumulation of methylglyoxal-modified proteins[11], peptides[12], and nucleotides[9], again raising the question about the true molecular function of DJ-1.

Heremans et al.[13] recently reported that DJ-1 prevents acylation of various cellular amines by the glycolytic metabolite 1,3-bisphosphoglycerate (1,3-BPG), an acyclic mixed anhydride of 3-phosphoglyceric and phosphoric acids. 1,3-BPG has been known to acylate lysine side chains in proteins resulting in 3-phosphoglyceroyl-lysine (pgK) modifications[14]. However, since DJ-1 neither uses 1,3-BPG as a common substrate, nor repairs pgK residues in modified proteins[13], it was suggested that the true substrate of DJ-1 is an unknown reactive intermediate product of 1,3-BPG decomposition. This intermediate product has not been identified, but based on

several kinetic experiments it was proposed that 1,3-BPG undergoes a slow reaction of cyclization facilitated by a loss of phosphate (Fig. 1a)[13]. The resulting molecule corresponds to a cyclic mixed 3-phosphoglyceric anhydride (cPGA) that is likely to be a highly reactive electrophile and acylating agent. Several lines of evidence indicate that DJ-1 and its homologs inactivate cPGA by hydrolysis explaining how DJ-1 prevents acylation of biomolecules by 1,3-BPG[13].

Indiscriminate reactions of cPGA (or another reactive intermediate of 1,3-BPG decomposition) with cellular components are potentially harmful and may exacerbate the effects of oxidative stress. Therefore, the true mechanism of cytoprotection by DJ-1 may be based on its unique ability to efficiently destroy cPGA. However, due to its presumed transient nature, the existence of cPGA has not been demonstrated by any of the standard spectroscopic techniques, and it is not clear whether or how it can be prepared and studied. Therefore, a direct demonstration and quantitative characterization of DJ-1

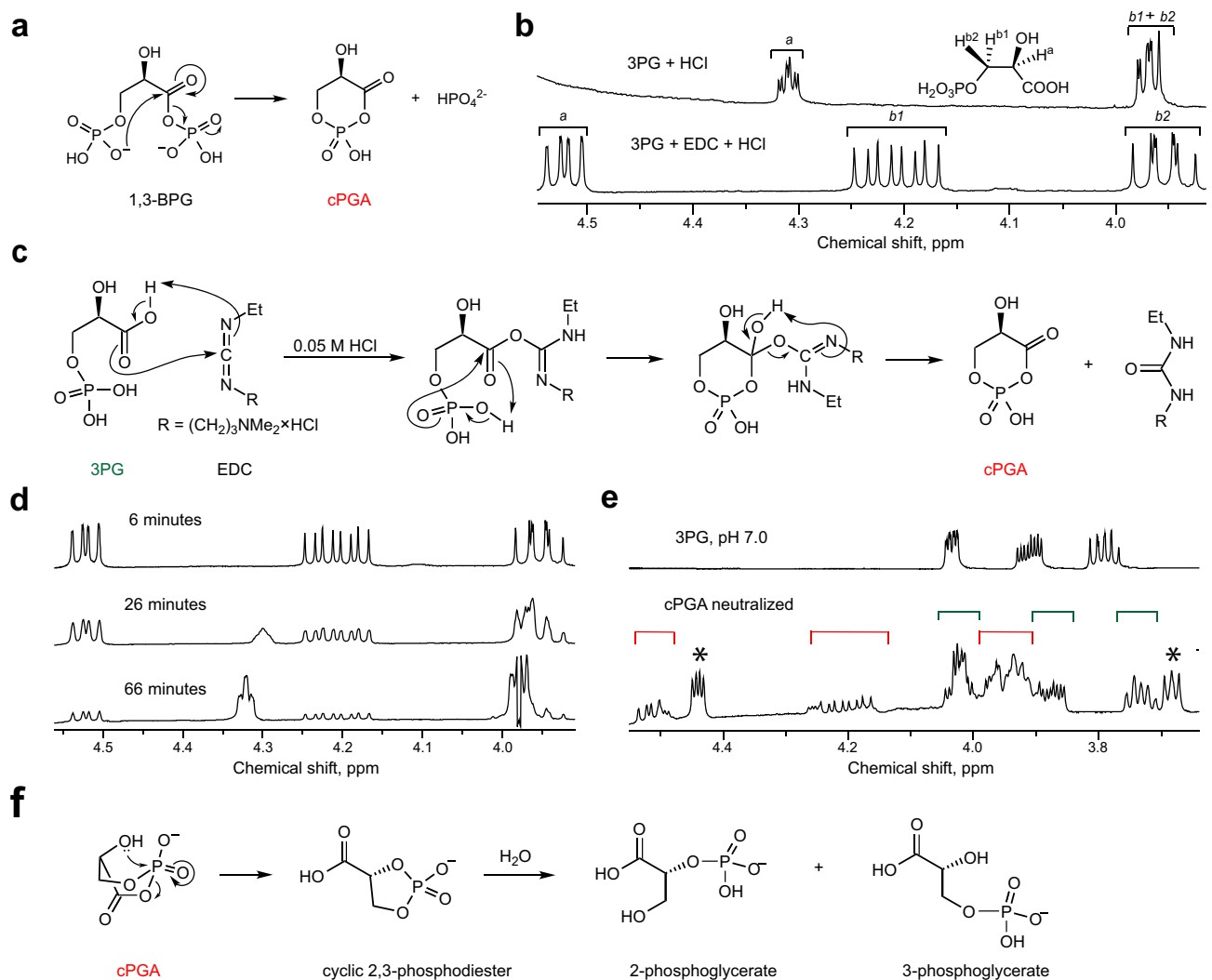

**Fig. 1 | Synthesis and decomposition of cPGA. a** Proposed mechanism of cPGA formation from 1,3-BPG. **b** In the presence of equimolar amounts of EDC and HCl, 3PG is immediately and completely converted to another molecule. [1]H-NMR spectra of 50 mM 3PG, or 3PG + EDC (50 mM each) in $D_2O$ supplemented with 50 mM HCl were recorded immediately after mixing of all reagents. Signals from the $H^a$, $H^{b1}$, and $H^{b2}$ protons are labeled as a, b1 and b2. **c** Proposed mechanism of formation of cPGA in reaction of 3PG with EDC. **d** A series of [1]H-NMR spectra of a mixture of 3PG, EDC and HCl (50 mM each) were recorded at indicated times after mixing of all reagents to illustrate the gradual spontaneous decomposition of cPGA back to 3PG. **e** 3PG, EDC, and HCl (50 mM each) were mixed in $D_2O$, incubated for

1 min, neutralized with 50 mM NaOH, and analyzed by [1]H-NMR. Signals of 3PG and cPGA are labeled with green and red square brackets, respectively. Signals that likely correspond to 2,3-phosphodiester of glyceric acid are labeled with asterisk. The [1]H-NMR spectrum of 3PG at pH 7.0 is shown for reference: a complex splitting pattern and peak broadening suggest that at neutral conditions 3PG is present in at least two different forms (e.g., monomer-dimer) that exist in quick equilibrium with each other. **f** Proposed mechanism of cPGA conversion into cyclic 2,3-phosphodiester at neutral pH. All NMR spectra are representative of at least 5 independent experiments.

enzymatic activity towards cPGA has not been achieved. From a broader perspective, if cPGA is indeed a previously unrecognized, ubiquitous, and toxic secondary metabolite, reliable protocols for its synthesis and quantitation should be established to facilitate research in this direction.

In this work, we synthesize cPGA and characterize it as an unstable electrophile that reacts rapidly with thiols and amines. Using synthesized cPGA as a substrate, we characterize human DJ-1 and its *E.coli* homolog YajL as highly efficient cPGA hydrolases. We demonstrate that endogenous cPGA hydrolase activity of DJ-1 in crude protein extracts is both necessary and sufficient to protect proteins from acylation by cPGA. Therefore, we conclude that the protective function of DJ-1 is due to its unique ability to inactivate by hydrolysis a highly reactive cyclic 3-phosphoglyceric anhydride that forms spontaneously in all cells undergoing glycolysis.

## Results

### Reaction of 3-phosphoglyceric acid with EDC produces cPGA

To raise antibodies against 3-phosphoglyceroyl lysine, we attempted activation of 3-phosphoglyceric acid (3PG) by EDC/NHS for subsequent coupling to KLH protein[14]. However, based on low amount of phosphate released from modified KLH by calf intestinal phosphatase, we found this protocol to be inefficient. In search of possible explanations for the inefficient modification, we studied the reaction of 3PG with EDC by [1]H-NMR. The [1]H-NMR spectrum of 3PG in acidic conditions (50 mM HCl) shows two signals: a triplet of doublets at 4.32 ppm corresponding to the $H^a$ proton signal splitting on diastereotopic $H^{b1}$ and $H^{b2}$ protons and a complex multiplet at 3.97 ppm due to splitting of the $H^{b1}$ and $H^{b2}$ proton signals on a phosphorus and $H^a$ (Fig. 1b). When an equimolar amount of EDC was added to 3PG in the presence of 50 mM HCl, proton signals corresponding to 3PG disappeared, and a new molecule with signals from three non-equivalent protons at 3.95, 4.2 and 4.52 ppm emerged in the 5 min required to record an NMR spectrum (Fig. 1b). At the same time, EDC was completely converted to its urea-like derivative (Supplementary Fig. 1), indicating that the dehydration of 3PG by EDC was complete and occurred with a very high yield. The [1]H-NMR spectrum of the product resulting from 3PG corresponds to a cyclic molecule because the $H^{b1}$ and $H^{b2}$ protons with nearly identical chemical shifts in an acyclic molecule of 3PG can give rise to two signals with very different chemical shifts only if they occupy non-equivalent equatorial and axial positions, for example within a six-membered ring. All three multiplet signals from the product agree with the chemical structure of cPGA: the doublet of doublets from the $H^a$ proton around 4.52 ppm is due to splitting on the $H^{b1}$ and $H^{b2}$ protons, while signals from the non-equivalent $H^{b1}$ and $H^{b2}$ protons display the expected 8 peaks due to splitting on adjacent hydrogen, $H^a$ and phosphorus (Fig. 1b).

We established the chemical identity of this product as cPGA beyond reasonable doubt by studying its decomposition, analyzing its reactions with thiols, amines, and proteins, and demonstrating that DJ-1 catalyzes its hydrolysis (see below). To explain the quick formation of cPGA from 3PG, we suggest that EDC-assisted cyclization proceeds via the anticipated formation of an O-acylisourea derivative followed by an immediate intramolecular nucleophilic displacement of the urea-like product by the terminal phosphate (Fig. 1c). When EDC was in a slight excess of 3PG, we observed that EDC signals were present but progressively declined for ~30 min after the start of reaction, while cPGA signals remained steady and declined only after EDC disappeared completely (Supplementary Fig. 2). These results confirm 1:1 stoichiometry between 3PG and EDC and suggest that after initial complete conversion of 3PG, the resulting cPGA is hydrolyzed back to 3PG which, in turn, reacts with EDC again to maintain a stable level of cPGA.

Monitoring the reaction by NMR revealed that cPGA produced from an equimolar mixture of 3PG and EDC decomposes back to 3PG in approximately an hour with no other intermediate products detectable in the [1]H-NMR spectrum (Fig. 1d). Since Heremans et al. [13] demonstrated that 2,3-cyclic phosphodiester is one of the products of reaction of 3PG with EDC, we hypothesized that cPGA can form at the different pH levels used by Heremans et al. (pH 5.5) and in our study (pH ~2.5) but may subsequently decompose via different pH-dependent mechanisms. To verify this hypothesis, we synthesized cPGA at pH 2.5, neutralized the reaction, and monitored the decomposition by [1]H-NMR. We observed formation of an additional product that is neither 3PG nor cPGA (Fig. 1e and Supplementary Fig. 3) but which gradually hydrolyses to 3PG and possibly to 2PG. We tentatively identified this product as 2,3-cyclic phosphodiester of glyceric acid and explain its predominant formation at pH 5.5 by an intramolecular nucleophilic attack involving a well-positioned 2-hydroxyl in boat conformation (Fig. 1f). In contrast, under the more acidic pH 2.5 used in our experiments, a smaller fraction of 2-hydroxyl would be deprotonated and inclined to nucleophilic attack so that a larger fraction of cPGA would be hydrolyzed directly to 3PG bypassing the formation of 2,3-cyclic phosphodiester.

### Characterization of cPGA as an acylating electrophile

Since cPGA did not show any significant optical absorbance, we designed a simple method for its quantitation based on its reaction with the thiol group of *N*-acetylcysteine (NAC). If we had indeed obtained cPGA in the above experiment, the reaction with NAC should quickly open its ring and produce a thioester (Fig. 2a), a member of a class of relatively stable compounds that can be quantified through characteristic optical absorbance around 235 nm. We found that cPGA reacted readily with NAC, isolated the product with an overall yield of 74% and identified it as a thioester by [1]H-NMR (Supplementary Fig. 4). An optical absorbance spectrum of the pure thioester showed a single peak at 235 nm with a corresponding molar extinction coefficient of $3.0\,mM^{-1}cm^{-1}$ (Fig. 2b), presenting a simple method to detect and quantify cPGA via the reaction with NAC.

Next, we studied the reaction kinetics of cPGA with NAC by monitoring the formation of thioester at 235 nm. We found that at concentrations of NAC of 1 mM or lower, formation of thioester takes several minutes at pH 7.0 (Fig. 2c) so that a significant fraction of cPGA is likely to escape detection/quantitation due to competing reactions of spontaneous decomposition. Therefore, we optimized derivatization conditions to allow quantification of cPGA by an end-point assay (Supplementary Fig. 5) and established a linear dependence of optical absorbance at 235 nm on cPGA concentration in the range of 0.1–1 mM (Fig. 2d). Comparing the slope of this calibration curve ($2.63\,mM^{-1}cm^{-1}$) with the extinction coefficient of pure thioester ($3.0\,mM^{-1}cm^{-1}$) we concluded that ~90% of the 3PG used to synthesize cPGA could be recovered as a thioester, attesting to high efficiencies of both cPGA synthesis and its subsequent derivatization with NAC. Next, we used the thioester-based end-point assay to investigate the stability of cPGA under the acidic conditions (pH ~2.5) used to produce cPGA and at neutral pH (50 mM phosphate buffer, pH 7.0). Under acidic conditions, cPGA is relatively stable at both room temperature and 0 °C with half-lives of 12 and 43 min, respectively. Conversely, cPGA half-life at neutral pH is only 3.9 min at room temperature and 19 min at 0 °C (Fig. 2e).

To quantify the reactivity of cPGA towards NAC, we determined a second-order rate constant for this reaction. We recorded kinetic curves by monitoring the optical absorbance at 235 nm for five different concentrations of cPGA reacting with 5 mM of NAC at pH 7.0 and fit them globally using KinTek software (Fig. 2f). We obtained excellent fits and derived a value of $7.1\,M^{-1}s^{-1}$ for the rate constant of cPGA reaction with NAC at pH 7.0 and 25 °C. To put this reactivity into physiological perspective, we repeated the experiment at 37 °C, pH 7.4, and 10 mM of NAC (to mimic the entirety of cytoplasmic thiols). Under these conditions, the half-reaction time is a few seconds (Fig. 2g), much slower than the half-time measured for the

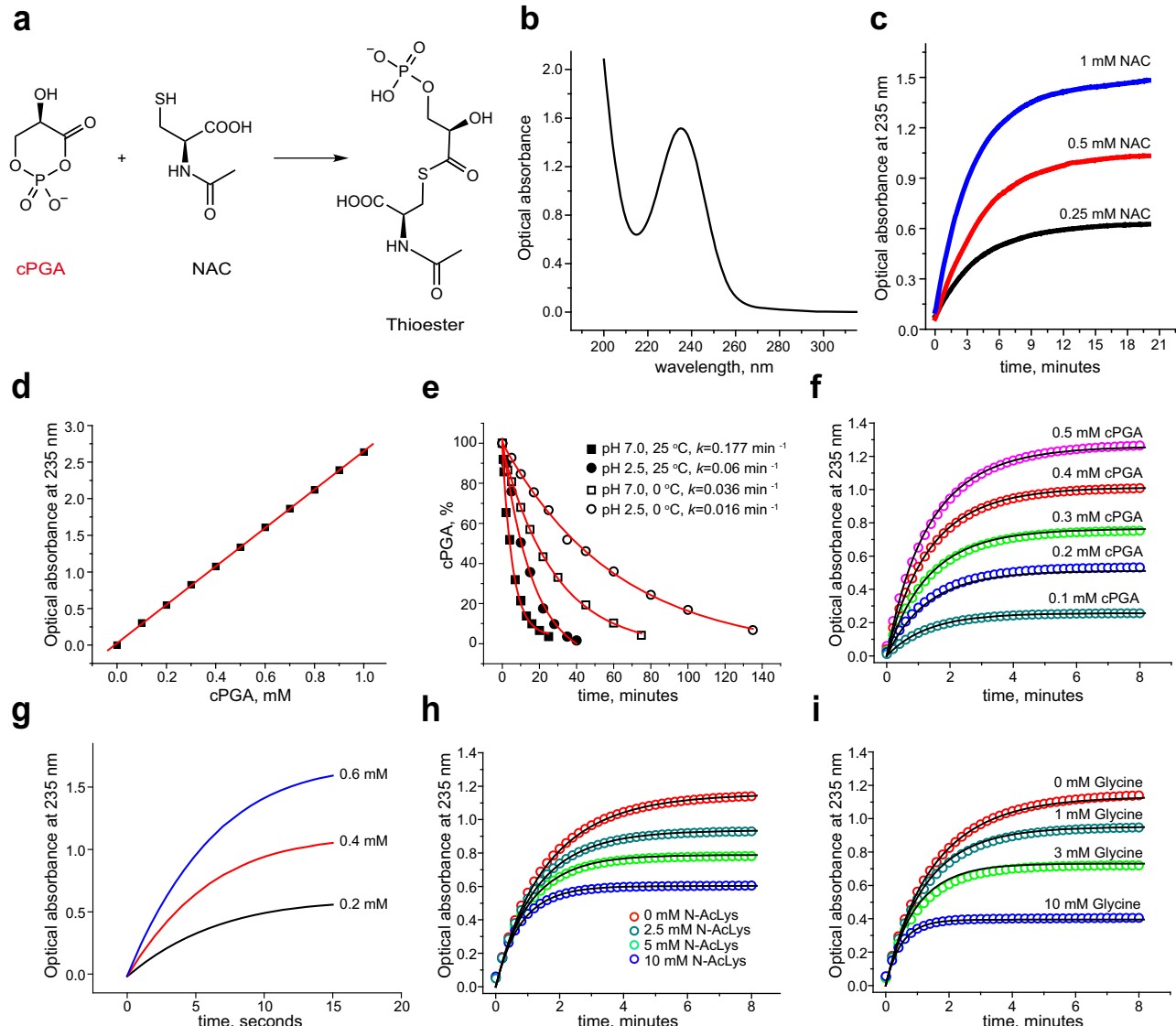

**Fig. 2 | cPGA is a highly reactive electrophile.** Unless otherwise indicated, all experiments were conducted in 50 mM sodium phosphate buffer (pH 7.0) at 25 °C. **a** Scheme for thioester formation in the reaction of cPGA with NAC. **b** UV absorbance spectrum of 0.5 mM of pure thioester. **c** Kinetics of thioester formation in the reaction of 0.75 mM cPGA with indicated concentrations of NAC was monitored by optical absorbance at 235 nm. **d** A standard curve for cPGA was obtained by derivatization with NAC. The linear fit to the data is shown as a red line. **e** Kinetics of spontaneous decomposition of cPGA at 0 and 25 °C in neutral and acidic conditions. cPGA concentrations were measured at indicated time points using the thioester-based end-point assay. Exponential decay fits are shown as red lines. **f** cPGA at indicated concentrations was allowed to react with 1.5 mM of NAC. The data were fit globally (solid lines) to derive the rate constant. **g** Kinetics of formation of thioester in the reaction of cPGA (0.2, 0.4, and 0.6 mM) with 10 mM NAC at pH 7.4 and 37 °C to mimic physiological conditions. Note that the reaction is essentially over within 10 s. **h** and **i** Reaction kinetics of NAC (5 mM) and cPGA (0.5 mM) in the presence of indicated concentrations of $N^{\alpha}$-acetyllysine (**h**) and glycine (**i**) as competitors. Global fits (solid black lines) were performed to derive rate constants of reaction of cPGA with $N^{\alpha}$-acetyllysine and glycine. Results shown in (**c**–**e** and **g**) are representative of 3–5 independent experiments. Results shown in (**f, h,** and **i**) represent one of the two independent series of experiments that showed the same results. Source data are provided as a Source Data file.

spontaneous decomposition of cPGA (Fig. 2e). These data suggest that nearly all the cPGA produced in a cell will acylate intracellular nucleophiles unless destroyed by an enzyme(s).

To characterize the reactivity of cPGA towards amines, we used glycine and $N^{\alpha}$-acetyllysine as competitors in the reaction of cPGA with NAC and obtained rate constants by globally fitting a series of kinetic curves (Fig. 2h, i). At pH 7.0 and room temperature, the rate constants of reaction with cPGA are 2.6 and 1.0 $M^{-1}s^{-1}$ for glycine and $N^{\alpha}$-acetyllysine, respectively. These results indicate that the reactivity of $\alpha$-amino groups of amino acids towards cPGA is comparable to that of thiols and suggest that the accumulation of N-glyceroyl and N-phosphoglyceroyl adducts in DJ-1 deficient cells reported by Heremans

et al.[13] can indeed result from direct acylation reactions of primary amine metabolites by cPGA.

## DJ-1 is a cPGA hydrolase

To gain insights into possible DJ-1 enzymatic activity towards cPGA, we incubated 0.5 mM of cPGA in phosphate buffer with different concentrations of DJ-1 for 1 min and then quantified the remaining cPGA by converting it into a thioester. After 1 min incubation without DJ-1, approximately 85% of cPGA remains due to its relatively slow rate of spontaneous decomposition (Fig. 3a). DJ-1 reduced cPGA concentration in a dose-dependent manner, demonstrating a high enzymatic activity: 10 nM of DJ-1 was sufficient to convert ~0.2 mM of cPGA in one

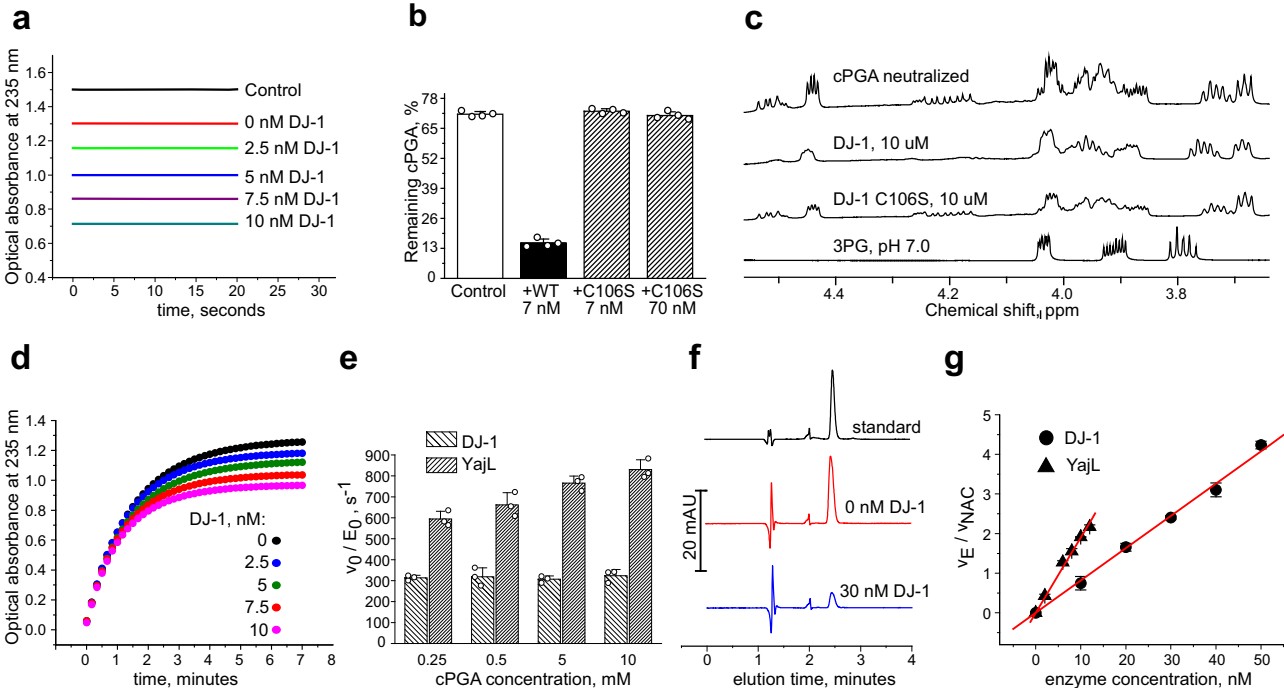

**Fig. 3 | Characterization of DJ-1 as cPGA hydrolase.** Unless otherwise indicated, all experiments were conducted in 50 mM sodium phosphate buffer (pH 7.0) at 25 °C. **a** cPGA (500 μM) was incubated with indicated concentrations of DJ-1 for 1 min at room temperature followed by an immediate conversion into NAC-thioester. Optical absorbance at 235 nm was recorded for 20 s to ensure the stability of signal after derivatization. **b** cPGA (500 μM) was incubated with DJ-1 or its C106S mutant at the indicated concentrations for 2 min followed by quantitation of the remaining cPGA by NAC assay (mean ± s.d. for $n = 4$ independent experiments). **c** $^1$H-NMR spectra of 50 mM cPGA neutralized with 50 mM $Na_2HPO_4$. DJ-1 or the C106S mutant were added immediately after neutralization. $^1$H-NMR spectrum of 3PG at pH 7.0 is shown for reference. **d** 0.5 mM of cPGA was allowed to react with 1.5 mM of NAC in the presence of DJ-1 at indicated concentrations. Thioester formation was monitored by optical absorbance at 235 nm. **e** Apparent catalytic constants ($v_0/E_0$) for DJ-1 and YajL were determined for the indicated concentrations of cPGA using end-point NAC assay (mean ± s.d. for $n = 3$ independent

experiments for each concentration of cPGA). **f** HPLC quantitation of NAC-thioester of phosphoglyceric acid. Synthetic thioester standard (50 μM) was used to quantify the amount of thioester formed in the reaction of 10 mM NAC with 100 μM cPGA (10 μM of cPGA were added 10 times every 3 min) in the presence or absence of 30 nM DJ-1. **g** To determine $k_{cat}/K_m$ of DJ-1 and YajL, enzymes at the indicated concentrations were allowed to compete with 10 mM of NAC for cPGA. The concentration of cPGA was kept in the low micromolar range to ensure a linear dependence of the rate of enzymatic conversion on the concentration of cPGA. The ratio of the enzymatic conversion rate of cPGA ($v_E$) to the reaction rate of cPGA with NAC ($v_{NAC}$) was determined by HPLC from amounts of accumulated thioester at the end of the experiment (mean ± s.e.m. for $n = 3$ independent experiments for each concentration of enzymes). Representative kinetic curves (**a**, **d**), NMR spectra (**c**) or HPLC traces (**f**) from three independent experiments are shown. Source data are provided as a Source Data file.

minute at room temperature (Fig. 3a). Mutation of the catalytic cysteine (C106S) in DJ-1 completely abolished this activity (Fig. 3b). To characterize further the enzymatic activity of DJ-1, we incubated cPGA with DJ-1 or with its C106S mutant and compared the resulting $^1$H-NMR spectra. We found that the C106S mutant did not affect the gradual transformation of cPGA into 2,3-cyclic phosphodiester, but incubation with wild-type DJ-1 resulted in a disappearance of cPGA signals (Fig. 3c) indicating that DJ-1 catalyzes the hydrolysis of cPGA to 3PG. This was further confirmed by NMR analysis of products of cPGA decomposition in the presence and in the absence of DJ-1 (Supplementary Fig. 6). The enzymatic activity of DJ-1 could also be observed in a continuous assay where DJ-1 was allowed to compete with NAC for cPGA: DJ-1 reduced both the yield and rate of thioester formation in a dose-dependent manner (Fig. 3d).

To determine the enzymatic parameters of cPGA hydrolysis catalyzed by DJ-1, we varied cPGA concentration between 0.25 and 10 mM in the reaction mixture and provided enough DJ-1 to convert 10–20% of cPGA in 1 min. We quantified the remaining cPGA by post-kinetic derivatization with NAC and found that the apparent $k_{cat}$ of DJ-1 (calculated as $v_0/E_0$ - a ratio of initial rate of DJ-1 catalyzed hydrolysis to DJ-1 concentration) remained stable around 310 s$^{-1}$ within a wide range of 0.25–10 mM cPGA (Fig. 3e), providing a good estimate for $k_{cat}$. YajL, an *E.coli* homolog of DJ-1, behaved similarly to DJ-1 but showed a much higher apparent $k_{cat}$ of up to ~800 s$^{-1}$ despite

the high structural conservation between the two proteins (41% sequence identity and main chain rmsd of 1.5 Å). Some concentration-dependent increase in the apparent $k_{cat}$ was observed for YajL, however, limitations of the NAC-based end-point assay did not allow testing concentrations of cPGA low enough to determine $K_m$ for either of the two enzymes. To estimate the catalytic efficiency of DJ-1 and YajL (equivalent to $k_{cat}/K_m$ for enzymes obeying Michaelis-Menten equation), we maintained a low concentration of cPGA by 10 sequential additions of 10 μM of cPGA to a reaction mixture containing 10 mM of buffered NAC in the presence or absence of DJ-1 or YajL. Under these conditions the rates of conversion of cPGA by both NAC and the enzymes are presumed to depend linearly on cPGA concentration, so that the apparent second-order rate constant ($k_{cat}/K_m$) could be determined by comparing the amounts of accumulated thioester in the presence and absence of competing enzymes (Supplementary Fig. 7). In these experiments, quantification of thioester was performed by HPLC (Fig. 3f) because slow oxidation of NAC over time contributed to absorbance at 235 nm and compromised spectrophotometric data. We derived room temperature $k_{cat}/K_m$ values of $5.9 \times 10^6$ M$^{-1}$s$^{-1}$ and $1.3 \times 10^7$ M$^{-1}$s$^{-1}$ for DJ-1 and YajL, respectively (Fig. 3g), indicating that both enzymes are highly efficient cPGA hydrolases. Calculated Michaelis constants for DJ-1 and YajL were similar at 51 and 62 μM, respectively.

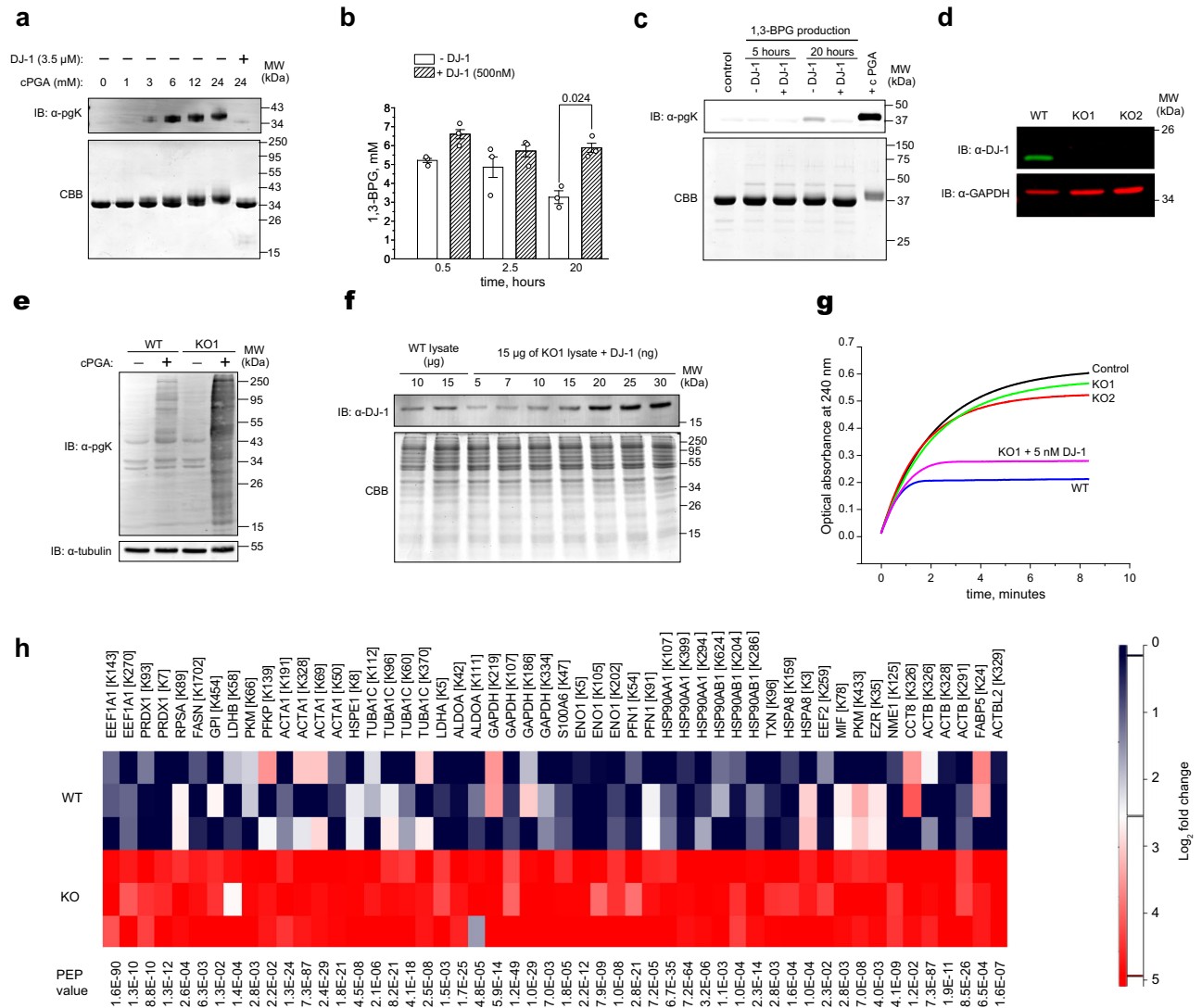

**Fig. 4 | DJ-1 protects proteins from acylation by 1,3-BPG and cPGA.**
**a** Immunoblot and Coomassie-stained gel of GAPDH (0.2 and 2 μg of GAPDH respectively) show that acylation by cPGA can be specifically detected by anti-3PG antibodies. **b** 1,3-BPG was produced continuously in the presence of 2.5 mg/mL of GAPDH that was later assayed for acylation by immunoblot. Concentration of 1,3-BPG was measured at indicated times and shown as the mean ± s.e.m. *P*-values were calculated using two-tailed paired Student's *t*-test (*n* = 3, independent experiments). **c** Acylation level of GAPDH in the presence of continuously produced 1,3-BPG was assessed by immunoblot. DJ-1 (500 nM) completely prevents acylation that is visible at 20 h of incubation. **d** Immunoblot analysis of DJ-1 expression in wild-type (WT) and DJ-1-null (KO1 and KO2) HCT116 cells. Bands corresponding to the molecular weight of DJ-1 are absent in both of the KO1 and KO2 clones. GAPDH was used as a loading control. **e** Immunoblot analysis of phosphoglyceroyl modifications in cytosol of WT and DJ-1-null cells (KO1) before and after treatment of cytosolic extracts with 0.5 mM cPGA for 20 min. Tubulin was used as a loading control. **f** To quantify DJ-1 in HCT116 cells, 10 and 15 μg of wild-type cytoplasmic

protein extract was immunoblotted along with 15 μg of extract from KO1 cells spiked with 5–30 ng of purified recombinant DJ-1. Coomassie brilliant blue (CBB) stained gel of the same samples is shown below. **g** Cytoplasmic protein extracts from wild-type HCT116 (0.1 mg/mL), KO1 (0.3 mg/mL), and KO2 (0.3 mg/mL) cells were added to a reaction of cPGA (0.3 mM) with NAC (1 mM). The formation of thioester was monitored by optical absorbance at 240 nm instead of 235 nm because of the strong absorbance of cell extracts. A strong inhibitory effect of WT extract on thioester accumulation could be recapitulated with KO1 lysate by adding 5 nM of DJ-1. **h** Quantitation of pgK modifications of cytoplasmic proteins from wild-type and DJ-1 knockout cells treated with cPGA. Protein extracts were treated by cPGA (1 mM of cPGA was added 5 times every 3 min) followed by proteolytic digestion and quantitation by LC/MS. Three technical replicates from a representative experiment are shown. Representative gels/blots (**a**, **c**–**f**) or kinetic curves (**g**) from three independent experiments are shown. Source data are provided as a Source Data file.

## DJ-1 prevents protein modification by cPGA

We acylated GAPDH with cPGA and used it to obtain rabbit polyclonal antibodies specific for pgK modifications (Supplementary Fig. 8 and Fig. 4a). In line with our initial observations, standard EDC/NHS protocol which involves 15 min pre-incubation of 3PG with EDC/NHS[14] was found to be less effective and likely resulted in modification of GAPDH by a mixture of 3PG and 2,3-phosphodiester of glyceric acid (Supplementary Fig. 8). Next, we used phosphoglycerate kinase and pyruvate kinase to couple phosphoenolpyruvate to the production of 1,3-BPG

(Fig. 4b). The concentration of 1,3-BPG reached 5 mM within 30 min and declined to 3 mM after 20 h of incubation. In the presence of DJ-1 the concentration of 1,3-BPG was slightly higher for the first 2.5 h, and much higher after 20 h of incubation (Fig. 4b). These results clearly indicate that DJ-1 does not catalyze the hydrolysis of 1,3-BPG in agreement with the previous study[13]. Higher levels of 1,3-BPG after 20 h of incubation in the presence of DJ-1 are likely due to enzymatic activity of DJ-1 that protects enzymes from inactivation by cPGA and regenerates spent 3PG. No significant acylation of GAPDH was detectable

after 5 h of continuous production of 1,3-BPG, however when incubation was extended to 20 h a significant increase in acylation level of GAPDH was detected (Fig. 4c). This increase in acylation was completely prevented by DJ-1 (Fig. 4c) suggesting that most of the acylation by 1,3-BPG is not direct but occurs via the formation of cPGA.

To assess the protective function of DJ-1 within a cellular context, we knocked out DJ-1 in HCT116 cancer cells and selected two knockout clones for further analysis (Fig. 4d). To qualitatively assess a possible protective role of DJ-1 against acylation by cPGA, we treated cytosolic extract from wild-type and knockout cells with 0.5 mM cPGA for 20 min. A clear difference in the level of cPGA-induced acylation was observed between knockout and wild-type cells suggesting that DJ-1 protects proteins from acylation by cPGA (Fig. 4e). Next, we quantified the endogenous DJ-1 in cytoplasmic protein extracts and evaluated its protective function against exogenously added cPGA. To construct a standard curve, known amounts of purified DJ-1 were spiked into cytoplasmic protein extracts from DJ-1 knockouts and immunoblotted alongside samples from wild-type cells. We found that DJ-1 is present at a level of 0.9 ng per $\mu$g of cytoplasmic proteins (Fig. 4f) which is high enough to demonstrate and quantify cPGA hydrolase activity in crude cytoplasmic protein extracts using the NAC competition assay. For example, cytoplasmic extract diluted to a concentration of 0.1 mg of protein per mL will contain about 5 nM of DJ-1, which is sufficient to compete with NAC for cPGA (Fig. 3d). We found that protein extract from wild-type HCT116 cells at 0.1 mg/mL strongly inhibited the accumulation of thioester in the reaction of 0.3 mM cPGA with 1 mM of NAC (Fig. 4g), suggesting that endogenous enzymatic activity destroys most cPGA before it can react with NAC. In contrast, protein extracts from both DJ-1 knockout clones had little to no effect on accumulation of thioester even at the much higher concentration of 0.3 mg/mL, indicating that DJ-1 is largely responsible for inactivation of cPGA in crude cytoplasmic protein extracts of HCT116 cells (Fig. 4g). The effect of protein extract from the wild-type HCT116 cells on thioester accumulation could be mimicked by adding 5 nM of DJ-1 to the cytoplasmic extract from KO1 clone, confirming that the observed effects can be explained entirely by the absence or presence of DJ-1 (Fig. 4g).

Next, we quantified and compared acylation of proteins by cPGA in cytoplasmic protein extracts of wild-type and DJ-1 knockout HCT116 cells. Label-free quantitation by nano LC-MS/MS revealed 54 phosphoglyceroyl-modified peptides from 30 different proteins in tryptic digests of cytoplasmic protein extracts of DJ-1 knockout cells treated with cPGA (Fig. 4h, Supplementary Fig. 9 and Supplementary Data 1). Most of the modified peptides were from proteins that are presumed to be highly abundant in the cytoplasm, such as cytoskeletal proteins and glycolytic enzymes, suggesting that cPGA acylates proteins indiscriminately. After averaging all the experiments, all 54 phosphoglyceroyl-modified peptides were found to be more abundant in samples derived from the knockout cells. On average, phosphoglyceroyl-modified peptides were 6.7 times more abundant in knockout samples, indicating that DJ-1 is required for effective protection of cytoplasmic proteins from acylation by cPGA.

## Discussion

Acylation of lysine residues in proteins by 1,3-BPG was reported by Moellering and Cravatt a decade ago[14]. The authors presumed that the electrophilic 1,3-BPG acylates proteins in direct reactions but several lines of evidence suggest that most of the acylation by 1,3-BPG is not direct but mediated by a reactive intermediate. First, various acyclic acylphosphates have been studied extensively and found to be relatively unreactive towards amines[15] and even thiols[16]. Second, DJ-1 can prevent acylation of small molecule amines and proteins in vitro without destroying 1,3-BPG[13]. Third, small molecule nucleophiles (cysteamine, ammonia) quench acylation with no effect on concentration of 1,3-BPG suggesting that acylation occurs via a rate-determining step of reactive intermediate formation[13]. Finally, the

many-fold increase in acylation levels of proteins and metabolites in cells and organisms lacking DJ-1[13] cannot be clearly explained by assuming that most of the glyceroyl and phosphoglyceroyl modifications result from direct reactions of 1,3-BPG with proteins and metabolites. The slow spontaneous formation of highly reactive cPGA from 1,3-BPG (Fig. 1a) suggested by Heremans et al.[13] reconciles these and other prior findings, but most importantly, it introduces cPGA as a probable, unrecognized metabolic byproduct of glycolysis with a high potential to indiscriminately acylate cellular nucleophiles. The very existence of a highly conserved subfamily of DJ-1 proteins with unique cPGA hydrolase activity confirmed and quantified in this study, attests to the damaging potential of cPGA and likelihood of its direct participation in various metabolic disorders.

Using acidic conditions (50 mM HCl) to both accelerate the reaction of 3PG with EDC and inhibit spontaneous decomposition of cPGA allows the production of cPGA with high yield while simultaneously destroying traces of unreacted EDC. As with 1,3-BPG (Fig. 1a), 3-phosphate is directly involved in an intramolecular substitution reaction, however since the isourea derivative of EDC is a much better leaving group than the phosphate, cyclization is instantaneous (Fig. 1b, c). As expected, cPGA is highly unstable with a half-life of ~4 min which is similar to the ~8 min half-life estimated by steady-state approximation of the kinetic profile of cPGA during continuous production of 1,3-BPG[13]. The major route of cPGA degradation at neutral pH is not direct hydrolysis of cyclic acylphosphate but rather an intramolecular attack by 2-hydroxyl resulting in a 2,3-cyclic phosphodiester of glyceric acid (Fig. 1e). Despite this intrinsic instability, we demonstrated that cPGA can be produced in situ, quantified by derivatization with NAC (Fig. 2d) and used in various experiments. For example, while standard protocols of derivatization of 3PG with EDC/NHS are unlikely to work well due to quick cPGA formation and degradation, cPGA itself can be prepared from 3PG and used as an acylating agent to prepare thioesters (Fig. 2b), amides and protein conjugates of 3PG (Supplementary Fig. 8).

The quantitative characterization of cPGA reactivity reported in this work (Fig. 2f–i) suggests that in addition to acylating cellular amines, cPGA will readily acylate thiol groups, for example the sulfhydryl group of glutathione or highly nucleophilic cysteines in active centers of enzymes. However, the resulting thioesters are much less likely to be detected by tandem mass spectrometry due to their intrinsic instability[17], therefore alternative methods may need to be employed to assess this type of cellular damage accurately. In general, the rate constants of cPGA reaction with thiols and amines are quite high and suggest that nearly all cPGA formed in the cytoplasm from 1,3-BPG will acylate cellular nucleophiles before it degrades spontaneously. Such a high reactivity implies that cPGA must be scavenged quickly after its spontaneous formation, demanding a very high catalytic efficiency from a scavenging enzyme.

The high catalytic efficiency of cPGA hydrolysis by DJ-1 ($5.9 \times 10^6$ M$^{-1}$s$^{-1}$ at room temperature and likely higher at 37 °C) strongly supports the idea that this catalytic activity represents its true enzymatic function. In particular, the $k_{cat}/K_m$ values for cPGA hydrolysis by DJ-1 are more than $10^4$ times higher than the $k_{cat}/K_m$ values for its methylglyoxalase activity (200–500 M$^{-1}$s$^{-1}$)[7,9,18]. Clearly, the methylglyoxalase activity of DJ-1 is unlikely to play any significant physiological role given the ubiquitous presence of the vastly more efficient Glyoxalase I with $k_{cat}/K_m$ of ~$10^7$ M$^{-1}$s$^{-1}$, and this assumption has been borne out in several reports[6,11,13]. Instead, our experiments demonstrate that the removal of DJ-1 completely abolishes the cPGA hydrolase activity of crude protein extract from HCT116 cells (Fig. 4g) and leaves cytoplasmic proteins unprotected from acylation by cPGA (Fig. 4h) suggesting that DJ-1 is indispensable for inactivation of cPGA.

Since mutations in DJ-1 are associated with an early onset of Parkinson's disease, slowly accumulating damage through cPGA provides a possible cause for the premature death of dopaminergic neurons in

affected individuals. Although the mechanisms of cPGA neurotoxicity are yet to be elucidated, several possibilities based on its chemical reactivity should be considered. Indiscriminate acylation of proteins by cPGA inactivates lysine's positive charge and creates a long, rather hydrophobic chain. Therefore, uncontrolled acylation by cPGA in dopaminergic neurons may trigger protein aggregation that leads to Lewy body formation and cell death. Indeed, the recent report describing a possible causative link between the loss of DJ-1 and Lewy body accumulation[19] would support this idea. A more specific mechanism of toxicity may involve reactions of cPGA with highly nucleophilic amino acids in the active sites of enzymes, including those responsible for protection against oxidative damage. Because DJ-1's catalytic cysteine is redox-sensitive and its oxidation causes inactivation of the enzyme[20,21], oxidative stress is likely to compound the damage from cPGA even in individuals with wild-type DJ-1. Indeed, several lines of evidence link oxidative stress and metabolic flux through glycolysis to age-related neuronal damage: accumulation of oxidized DJ-1 in the aging brain[22], implication of phosphoglycerate kinase-1 in juvenile Parkinsonism[23], and high levels of oxidative damage in a DJ-1-deficient dopamine-producing cell line[24]. Finally, neurotoxicity could also be caused by an inability of affected neurons to efficiently process a variety of pgK and gK adducts of primary amines which may lead to lysosomal dysfunction, a pathology that has been reported for a DJ-1-deficient cell line[24].

In conclusion, our data demonstrate that DJ-1 possesses a highly efficient cPGA hydrolase activity that is many orders of magnitude higher than any of its other reported enzymatic activities. This activity is essential to prevent acylation of biomolecules by cPGA, providing a potential mechanistic link to cytoprotective effects of DJ-1. This work establishes a firm foundation upon which the role of cPGA hydrolase activity of DJ-1 in neuroprotection can be further investigated.

## Methods

### Chemicals and antibodies
Unless indicated otherwise, all chemicals were obtained from Sigma–Aldrich. Alexa Fluor 680 α-GAPDH, 680RD α-rabbit IgG and 800CW α-mouse IgG antibodies were from Abcam. Anti-acetylated tubulin antibodies were from Santa Cruz Biotechnology. Rabbit polyclonal antibodies against human DJ-1[6] and rabbit α-pgK antibodies were produced in the National Center for Biotechnology, Astana, Kazakhstan.

### Preparation of cPGA
Solid EDC was dissolved in freshly prepared, ice-cold solution of 3PG and HCl to obtain a solution with 50 mM final concentrations of all reactants. After incubation for 5 min on ice, the solution was flash-frozen in 50–500 µL aliquots. For $^1$H-NMR experiments, reactants were mixed in $D_2O$ at room temperature, and spectra were recorded immediately and every 10 min thereafter.

### Synthesis of *N*-acetylcysteine thioester of 3-phosphoglyceric acid
3PG (75 µmol, 375 µL) and HCl (75 µmol, 12.5 µL) were mixed with 737 µL of cold $H_2O$ on ice and incubated for 5 min, then EDC (75 µmol, 375 µL) was added. The reaction was quickly mixed and incubated on ice for 3 min after which 140 µL of *N*-acetylcysteine in NaOH (98 µmol N-Acetylcysteine in 176 µmol NaOH) was added. The resulting thioester was purified in 3 batches by FPLC on a 1 mL Resource Q anion exchange column using gradient elution from 0 to 60 mM HCl. The collected fractions were concentrated on a rotary evaporator overnight at room temperature to afford 18.4 mg of pure thioester (74% yield). $^1$H-NMR (500 MHz, $D_2O$) δ 4.48 (dd, 1H, $J$ = 7.3, 4.8 Hz, H4), 4.41–4.32 (m, 1H, H2), 4.06–3.92 (m, 2H, H1′ and H1″), 3.36 (dd, 1H, $J$ = 14.3, 4.8 Hz, H3′), 3.16 (dd, 1H, $J$ = 14.3, 7.3 Hz, H3″), 1.85 (s, 3H, H5); UV/VIS: $\lambda_{max}$ 235 nm; HRMS (m/z): [M-H]$^-$ calcd. for $C_8H_{13}NO_9PS^-$, 330.0054; found, 330.0050

### NAC-based spectrophotometric assay for cPGA
Samples of cPGA in 50 mM phosphate buffer (pH 7.0) were quickly mixed with NAC and NaOH (5 and 20 mM final concentrations, respectively) and incubated for 1 min after which HCl was added to a final concentration of 60 mM. $OD_{235}$ values were recorded for 20 s on a Shimadzu UV-2600i spectrophotometer and averaged. Concentrations of cPGA were calculated using $\varepsilon_{235} = 3.0$ mM$^{-1}$cm$^{-1}$ after accounting for ~7% dilution due to addition of NAC, NaOH, and HCl.

### Stability and reactivity of cPGA towards thiols and amines
For stability assays, aliquots of cPGA were quickly thawed and diluted with either 50 mM phosphate buffer (pH 7.0) or 50 mM HCl to a concentration of 0.5 mM. Aliquots (50–100 µL) were taken at appropriate time points to quantify the remaining cPGA by the NAC-based endpoint assay. The obtained data were fit with an exponential decay function. To determine rate constants, 2 mL reactions were set up in 50 mM sodium phosphate buffer (pH 7.0) containing all reactants except cPGA. To initiate the reaction, an aliquot of cPGA was thawed and immediately added to the reaction mixture. Kinetic data were collected for 5–10 min by monitoring optical absorbance at 235 nm on a Shimadzu UV-2600i spectrophotometer. To determine the rate constant of the cPGA-NAC reaction, absorbance data from 5 reactions with varying concentrations of cPGA were globally fit using KinTek global kinetic explorer software (version 11.01). The formation of thioester was simulated using $\varepsilon_{235} = 3.0$ mM$^{-1}$cm$^{-1}$, the initial concentrations of reactants, and a fixed rate constant of spontaneous cPGA decomposition. The model included a second-order one-step reaction of cPGA with NAC (for which the rate constant was extracted by globally fitting the model to a series of reaction kinetics) and a first-order cPGA decomposition for which the rate constant was determined in a separate experiment and fixed at 0.003 s$^{-1}$. Rate constants for cPGA reactions with glycine and $N^\alpha$-acetyllysine were obtained in a similar manner by including a corresponding reaction step into the model.

### Continuous production of 1,3-BPG
To achieve a high in situ concentration of 1,3-BPG, we adapted the procedure used for enzymatic synthesis of 1,3-BPG[13] for continuous production of 1,3-BPG over extended periods of time. Final concentrations of reagents were 12 mM HEPES pH 7.4, 25 mM 3PG, 15 mM ATP, 6 mM $MgCl_2$, 40 mM KCl, 40 µg/mL *E.coli* 3-phosphoglycerate kinase, 40 µg/mL rabbit muscle pyruvate kinase (Roche), 70 mM of neutralized phosphoenolpyruvate and 2.5 mg/mL of human GAPDH. GAPDH which was present in large excess overall other enzymes was later assayed for acylation by 3PG using immunoblot with rabbit polyclonal anti-pgK. The reaction started with 40 mM phosphoenolpyruvate and was incubated at room temperature for 15 min. After that, the remaining 30 mM of phosphoenolpyruvate were added and incubation continued for 24 h. To assess the involvement of cPGA in acylation, an identical sample with 0.5 µM of DJ-1 was set up and processed in parallel to the control experiment. To assess acylation of GAPDH by 1,3-BPG, aliquots were removed at 5 and 20 h and analyzed by immunoblot. To determine the concentration of 1,3-BPG, 10 µL aliquots were mixed with 10 µL of methanol to inactivate enzymes, followed by spectrophotometric quantitation of NADH in GAPDH-catalyzed conversion[13].

### Quantification of *N*-acetylcysteine thioester of 3-phosphoglyceric acid by HPLC
HPLC set up consisted of Thermo Scientific™ UltiMate™ 3000 UHPLC controlled by the Chromeleon 7.2 software and equipped with an autosampler (WPS-3000TSL), a column oven (TCC-3100), and a diode array detector (UV-vis DAD-3000RS). Samples were separated on Hypersil GOLD C18 columns (150 × 2.1 mm, 1.9 µm, Thermo Scientific). A mixture of 20 mM phosphoric acid and acetonitrile (99:1 v/v) was

filtered through a 0.45-µm filter and used as a mobile phase for iso-cratic elution. Separation was performed at a flow rate of 0.3 mL/min at 35 °C. The volume of injection was 5 µL, and the absorbance of the effluent was recorded at 240 nm for a total run time of 6 min. The use of a diode array detector allowed absorption spectra of separate peaks to be retrieved in the 190–800 nm range after the run. $N$-acetylcysteine thioester of 3-phosphoglyceric acid eluted at 2.6–2.7 min.

## $^1$H-NMR spectroscopy
$^1$H spectra of samples prepared in $D_2O$ were obtained with JEOL ECA-500 FT NMR spectrometer at 500 MHz and 20 °C. $^1$H chemical shifts were referenced to the residual protons in $D_2O$.

## Anti-pgK antibody production
Human GAPDH (5 mg/mL in 0.1 M sodium phosphate buffer pH 7.6) was modified with 17 mM cPGA for 60 min followed by desalting into PBS. For immunization, the resulting protein was diluted to 0.6 mg/mL with PBS and mixed at 1:1 ratio with Freunds' adjuvant. This solution was injected subcutaneously (-0.3 mg of GAPDH) into two 4-month-old white New Zealand male rabbits (*Oryctolagus cuniculus)*. Booster immunizations were performed at 7, 14, 21, and 28 days followed by blood collection at 6 weeks post-inoculation. Antibodies were partially purified by two-step precipitation with ammonium sulfate (precipitate from 35% saturation was discarded and precipitate from 45% satura-tion was used further). Precipitated antibodies were collected by centrifugation, resuspended in 12 mL of PBS, and dialyzed against PBS plus 0.05% of sodium azide overnight. To remove antibodies recog-nizing unmodified GAPDH, a portion of antibody solution (-2 mL) was passed through a 1 mL Ni-NTA column loaded with 5 mg of His-tagged GAPDH. The production of anti-pgK was approved by the Institutional Animal Care and Use Committee (IACUC) of the National Center for Biotechnology (IRB00013497 National Center for Biotechnology IRB #3). The authors assert that all experiments were performed in accordance with relevant guidelines and regulations. All research work with laboratory animals was performed in accordance with generally accepted ethical standards and comply with the rules adopted by the European Convention for the Protection of Vertebrate Animals Used for Research and Other Scientific Purposes.

## Expression and purification of DJ-1, PGK, YajL and GAPDH
His-tagged, full-length human DJ-1 was expressed and purified on a Ni-NTA column[25] followed by the removal of the His-tag by TEV protease. The resulting sample was desalted on a PD-10 column into a storage buffer (20 mM Tris, 0.5 M NaCl, 1 mM 2-mercaptoethanol), passed through a fresh Ni-NTA column and frozen in 50 µL aliquots. Open reading frames were amplified by PCR from *E.coli* genomic DNA using primers:

5′-AACCATGGGAATGTCTGTAATTAAGATGACCGATC-3′ and
5′-AAGTCGACTTACTTCTTAGCGCGCTCTTCGAGCATC-3′      for 3-phosphoglycerate kinase and
5′-GCGCCATGGGAATGAGCGCATCGGC-3′ and
5′-GCGTCGACCTACTCGTAATAATTATAAA-3′ for YajL.

Amplified fragments were cloned into pHisParallel vector[26], expressed, and purified in the same manner as DJ-1. Full-length human GAPDH was cloned into pHisParallel vector[26], expressed, and purified in the same manner as DJ-1.

## Enzyme assays
To measure the apparent $k_{cat}$, freshly thawed aliquots of cPGA were immediately diluted either with 50 mM phosphate buffer (pH 7.0) or with 50 mM phosphate buffer containing 3–100 nM of DJ-1 or YajL. After one minute the concentration of the remaining cPGA was mea-sured by the spectrophotometric NAC assay. For concentrations of cPGA above 0.5 mM, reactions were quickly diluted to 0.5 mM of cPGA (initial concentration) before the NAC assay. For each cPGA

concentration measured, the enzyme concentration was adjusted by trial and error to ensure a 10–20% drop in absorbance at 235 nm compared to the negative control. To measure $k_{cat}/K_m$, 10 µL aliquots of 5 mM cPGA were repeatedly added to 5 mL of 50 mM phosphate buffer (pH 7.0) containing 10 mM of neutralized NAC and 0–50 nM of DJ-1 or YajL. In total, cPGA was added 10 times every 3 min to maintain a low concentration throughout the experiment. The amount of accumulated thioester was determined by HPLC using a synthetic standard for quantitation. For every enzyme concentration, the amount of cPGA enzymatically converted that is proportional to the rate of the enzyme-catalyzed reaction ($v_e$) was deduced by comparing thioester amounts to samples with no enzyme added ($v_{NAC}$). The ratios of $v_e/v_{NAC}$ were plotted versus corresponding enzyme concentrations, and the values of $k_{cat}/K_m$ were determined from the slope of the curve (Supplementary Fig. 7).

## Preparation of cytoplasmic protein extract
HCT−116 cells grown on 10 cm Petri dishes were washed twice with PBS and lysed in 1 mL of lysis buffer (10 mM HEPES, 60 mM KCl, 0.1% (v/v) NP-40, pH 7.6, protease inhibitors cocktail), gently vortexed for 1 min, incubated on ice for 5 min and centrifuged for 5 min at $5000 \times g$ and 4 °C. Benzonase (250 U) was added, the extract was incubated for 10 min on ice and desalted on a PD−10 column into 50 mM sodium phosphate buffer (pH 7.0). Protein concentration was quantified by BCA assay.

## Nano-LC−MS/MS analysis of peptides
A Dionex UltiMate 3000 RSLCnano System coupled to an Impact II QTOF tandem mass spectrometer (Bruker Daltonics) via a Captive-Spray nanoBooster ion source[27] was used for analysis of peptide samples. LC−MS/MS data were acquired using a Shotgun or data-dependent acquisition (DDA) method[28].

## Bioinformatics and statistical analysis
Detailed methods are described in Supplementary Information. Briefly, DDA raw data were processed using MaxQuant software (ver-sion 2.4.0.0) with Andromeda search engine[29], which is freely available for analysis of Shotgun proteomics experimental data. Label-free quantitation was based on peak areas reconstructed from MS1 pre-cursor ion intensities[30]. Additional data analysis was performed using the MaxQuant Viewer and the Perseus post-data capture package (version 2.0.10.0)[31]. The final matrix was subjected to the statistical ANOVA test with permutation-based false discovery rate of 0.05. Hierarchical clustering and scaled-to-interval normalization were car-ried out using filtered and $\log_2$-transformed data.

## Cell culture
HCT−116 cells were purchased from ATCC and maintained at 37 °C and 5% $CO_2$. Cells were cultured in high-glucose Dulbecco's modified Eagle's medium (Gibco) supplemented with 10% fetal bovine serum (Gibco), 2 mM glutamine, and antibiotics.

## CRISPR/Cas-mediated generation of DJ-1 knockout cell lines
DJ-1 knockout HCT116 cell lines were generated as described earlier[6]. In brief, exons 2 and 3 of DJ-1 gene were targeted to minimize the risk of off-target effects of selected guide RNAs. The sgRNA sequences were synthesized as short oligos and cloned into pX330-U6-Chimeric_BB-CBh-hSpCas9 plasmid (Addgene, #42230) under the U6 promoter using BbsI restriction sites. HCT116 cells were seeded into 6 well plate at a density of 400,000 cells per well. The next day the cells were transfected with 3 µg of expression construct and 1.5 µg of vector pPGKpuro (Addgene, #11349) using Lipofectamine 3000 (Invitrogen) according to manufacturer's instructions. Antibiotic selection medium containing 0.8 µg/mL puromycin was added to the cells 48 h post-transfection and transfected cells were selected using cloning rings.

Knockout clones were validated by genomic DNA sequencing and immunoblotting. Two knockout clones (KO1 and KO2) generated by different sgRNA sequences were selected for further work.

## Western blotting
Cell lysates were separated by 4–12% SDS-PAGE and transferred onto PVDF membranes. Blots were blocked overnight at 4 °C in 5% non-fat dry milk and 2% gelatin in TBS and 0.1% Tween-20. Blots were incubated in primary antibodies (1:1000 or 1:2000) for 1 h and with secondary antibodies (1:20,000) for 45 min at room temperature. Secondary antibody signal was measured using the Odyssey CLx infrared imaging system (LI-COR).

## Acylation of cytoplasmic proteins with cPGA for immunoblot detection of 3PG modifications
Wild-type and knockout HCT116 cells grown on 10 cm plates at ~80% confluency were scraped into 200 μL of 20 mM Hepes buffer pH 7.6 containing 1% of sodium deoxycholate and 50 mM KCl. After centrifugation at $21000 \times g$ for 20 min, the clarified lysate (~8 mg/mL of protein) was incubated with 0.5 mM cPGA for 20 min.

## Acylation of cytoplasmic proteins with cPGA and LC–MS/MS sample preparation
The cytosolic fraction of HCT116 cells in sodium phosphate buffer (50 mM, pH 7.0) at a concentration of 1.7 mg/mL was mixed with 1 mM of cPGA every three minutes for five times. Twenty minutes after final addition of cPGA solid urea was added to the sample to a final concentration of 8 M. Samples were reduced (5 mM DTT, 25 min, 56 °C), carbamidomethylated (15 mM IAA, 30 min, in the dark at room temperature), and diluted with 50 mM Tris–HCl to reach ≤5.5 M urea. Enzyme digestion was started by adding LysC (New England Biolabs, 1:50) and continued for twelve hours at 37 °C, followed by a five-hour incubation with trypsin-ultra (New England Biolabs, 1:50). The digested protein solution was adjusted to 0.1% TFA and 3 μg were desalted using a ZipTip-C$_{18}$ (Millipore) according to manufacturer's instructions. Peptide eluates were dried in vacuum concentrator and dissolved in 0.1% TFA.

## LC–MS/MS analysis
Acetonitrile, LC/MS grade (Fluka, #14261-1 L) and formic acid for mass spectrometry (Fluka, #94318-50ML-F) were used for preparation of buffers A and B for liquid chromatography. Buffer A: 0.1% formic acid; Buffer B: 90% acetonitrile/10% H$_2$O/0.1% formic acid. The LC–MS/MS analysis was conducted with a nanoflow HPLC system (Thermo Dionex Ultimate 3000, Thermo Scientific) equipped with Acclaim PepMap100 C18 pre-column (5 mm × 300 μm; 5 μm particles; Thermo Scientific, #160454) and Acclaim Pep-Map RSLC column (15 cm × 75 μm, C18, 2 μm, 100 Å particles; Thermo Scientific, #164534). The Dionex UltiMate 3000 RSLCnano System was coupled to an Impact II QTOF tandem mass spectrometer (Bruker Daltonics) via a CaptiveSpray nanoBooster ion source. Acetonitrile (R Chromasolv) for liquid chromatography (Sigma–Aldrich, #34881-2.5 L) was used with the CaptiveSpray nanoBooster system. Raw data were inspected and analyzed with the Bruker Compass DataAnalysis (version 4.3) software.

## Nano LC–MS/MS data acquisition
Degassed solvents were used for nanoLC (mobile phase (A): 0.1% (vol/vol) formic acid in water; organic phase (B) 0.1% (vol/vol) formic acid in acetonitrile). Peptide separations were performed at a flow rate 300 nl/min and 60 °C using linear gradient of 2–50% B for 90 min, 50–99% of B for 1 min, and 99% of B for 10 min followed by re-equilibration at 2% B for a total of 120 min. Ionization was achieved using CaptiveSpray nanoBooster source and a capillary voltage of 1000 V. The dry gas temperature was set at 150 °C with a flow of 3 L/min. The total cycle time was set at 3.0 s. The collision energy was adapted in the 25–52 eV range depending on the m/z value.

## Creating DDA method and analysis of raw data
The mass range of the MS scan was set between m/z 150 and 2200 in positive ion polarity mode. The spectra rate for MS was set at 4.00 Hz and for MS/MS at 8.00 Hz for low-intensity signals (threshold 1250 cts) and at 16.00 Hz for high-intensity signals (threshold 12500 cts). DDA raw data were processed using MaxQuant software (version 2.4.0.0) with the Andromeda search engine[29], which is freely available for analysis of Shotgun proteomics experimental data[32,33]. The false discovery rate (FDR) was set to 1% for both proteins and peptides. The permitted fragment mass deviation was set to 40 ppm, while the first search peptide tolerance was set at 0.07 Da. For quantitative studies, the MS/MS spectral searches against the Uniprot human database (UP000005640_9606, containing 82,492 protein counts) were performed using the Andromeda search engine. In addition to allowing cleavage at proline bonds and a maximum of two missed cleavages, the enzyme specificity was specified to be C-terminal to Arg and Lys. Carbamidomethylation of cysteine was chosen as a fixed modification, whereas methionine oxidation, N-terminal protein acetylation, lysine phosphoglyceroylation, and N-terminal phosphoglyceroylation were chosen as variable modifications (the last two variable modifications were added manually to the list of modifications). In quantification experiments, identifications were transferred to additional LC–MS/MS runs based on their masses and retention times (maximum deviance 0.7 min) using MaxQuant's "match between runs" feature[30]. Label-free techniques (LFQ) were used for quantifications[34] with at least one "razor peptide" and a minimum peptide ratio count of two. Additional data analysis was carried out using the MaxQuant Viewer and the Perseus (version 2.0.10.0) post-data capture package[31]. Reverse peptides and potential contaminants were removed from the matrix and two biological and three technical replicates were log$_2$ transformed. Imputation was used to fill in for missing data values after label-free protein quantification intensities were filtered for valid values with a minimum of 70% valid values per group. The final matrix was subjected to an ANOVA test with permutation-based FDR of 0.05. Hierarchical clustering and scaled-to-interval normalization were carried out using filtered and log$_2$-transformed data.

## Reporting summary
Further information on research design is available in the Nature Portfolio Reporting Summary linked to this article.

## Data availability
The mass spectrometry proteomics data have been deposited to the ProteomeXchange Consortium (http://proteomecentral. proteomexchange.org) via the iProX partner repository[35,36] with the dataset identifier PXD043887. Raw NMR spectra are available from the corresponding author upon request. The remaining data are available within the Article, Supplementary Information, Supplementary Data or Source Data file. Source data are provided with this paper.

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

## Acknowledgements

We thank the core facility at Nazarbayev University for help with NMR spectroscopy experiments, HPLC, oligonucleotide synthesis, and DNA sequencing. We thank Adilet Bekmagambetov for technical assistance. This work was supported by a Nazarbayev University CRP grant 091019CRP2121 to D.U. and a Nazarbayev University FDCRP grant 240919FD3910 to An.A.

## Author contributions

D.U. designed the research; Ai.A., A.Y., N.D., A.M., Z.S. A.K. and An.A. performed the experiments; Ai.A., A.K. and D.U. analyzed the data; D.U. wrote the paper with the other authors' help. D.U. directed the research.

## Competing interests

The authors declare no competing interests.
