## [Peer Review File · Nature Communications]

DJ-1 protects proteins from acylation by catalyzing the hydrolysis of highly reactive cyclic 3-phosphoglyceric anhydrideREVIEWER COMMENTS

Reviewer #1 (Remarks to the Author):

The authors report a kinetic and cellular study of DJ-1 activity against cyclic 1,3 phosphoglycerate anhydride (cPGA). This electrophile was previously proposed by Heremans et al (PMID: 35046029) to form spontaneously from 1,3 bisphosphoglycerate and to damage cellular nucleophiles. That same report showed that DJ-1 and its bacterial homolog YajL robustly defend proteins against this modification, which Heremans et al proposed was due to DJ-1's action on this proposed metabolite. In this report, the authors test this hypothesis by generating cPGA with a clever in vitro method that permits them to measure DJ-1 and YajL catalytic activity against this substrate. Confirming the prior proposal, they find that both proteins are highly efficient at converting cPGA to 3-phosphoglycerate, with k_{cat}/K_m values of 10^6 - 10^7 . These catalytic efficiencies are several orders of magnitude higher than previously reported DJ-1 methylglyoxalase values and support the contention that this is a physiologically significant activity for DJ-1. They then corroborate Heremans et al.'s results showing that DJ-1 and YajL strongly reduce the amount of glycerate modification on diverse cellular proteins. In all, this is an impressive study that succeeds in generating an unstable, reactive metabolite in sufficient quantity to assay DJ-1 activity in vitro. The results confirm prior reports but also advance the field by permitting in vitro study of this activity, which is likely to be important for understanding DJ-1's function. The manuscript is clearly written and well-illustrated with a high degree of experimental rigor. My principal reservation is that the extent that this clearly demonstrated molecular activity is responsible for the much-studied cytoprotective activities of DJ-1 is still unknown, and there are some statements in the manuscript that should be softened in light of this. This and some other points for the authors to consider are detailed below.

1) The title of the manuscript is "DJ-1 protects cells by catalyzing the hydrolysis of cyclic 3-phosphoglyceric anhydride - a highly reactive glycolytic byproduct", but there are no data in the paper that address cellular protection by DJ-1. To be clear, this reviewer thinks it is likely that the activity measured by the authors makes an important contribution to DJ-1 biology and the authors present data establishing protection of proteins from cPGA modification, but this is different from cytoprotection. This is an essential distinction because the degree to which this new activity is responsible for established aspects of DJ-1 cytoprotection (e.g. anti-oxidative stress, modulation of the PTEN/Akt, ASK1, and other pathways) is not known and likely will be an important direction for future research. I think it would suffice for the authors to state that DJ-1 protects proteins from damage by cyclic 1,3 phosphoglycerate, which is directly addressed by their data.

2) Related to (1), there are several places in the manuscript where the authors state that activity against cPGA is the "true molecular function of DJ-1". I find the data presented in this manuscript showing that DJ-1 is highly active against cPGA to be quite convincing, which is exciting and opens a new chapter in the study of DJ-1. However, statements of "true molecular function" in the manuscript are sweeping and should be softened or removed. This also affects the final paragraph of the Results section (line 273, p.9).

Like GAPDH (PMID: 20727968, 21895736), it seems likely that DJ-1 has multiple functions in cells, and multifunctionality (i.e. moonlighting) is a common theme in proteins (PMID: 29203708, 20144902). The GAPDH example is instructive: like DJ-1, it is an oligomeric cysteine-dependent protein that acts on triose phosphates and is subject to diverse post-translational modifications that alter its cellular function. The results presented by the authors are impressive enough that they do not require categorical statements of “true” function. This work provides a firm foundation upon which the role of this newly discovered DJ-1 activity can be further investigated.

3) P. 2 “However, no signaling pathways or binding partners have been unequivocally established for DJ-1, leaving the mechanism of neuroprotection by DJ-1 an open question”. I acknowledge that “unequivocally” allows room for debate, but there are several high-quality reports that connect DJ-1 to the PTEN/Akt pathway (PMID: 15766664, many others), the ASK1 pathway (PMID: 19293155, 20385180), and several other pathways. It would be fair to rephrase to state that no clear, consensus molecular mechanism for DJ-1 cytoprotection currently exists.

4) There are some typographical errors (e.g. neutralized in Fig. 1, etc) that should be corrected.

Reviewer #2 (Remarks to the Author):

DJ1/PARK7 mutations are the cause of autosomal recessive forms of early-onset Parkinson’s disease, but the molecular cause has remained elusive for a long time. In 2022, Heremans and colleagues found that this protein prevents acylation of metabolites and proteins by 1,3-bisphosphoglycerate. Evidence was presented that this acylation proceeds via cyclic-1,3-phosphoglycerate (referred to in the present work as cyclic-3-phosphoglyceric anhydride (cPGA). However, the enzymatic activity was never directly demonstrated and the cyclic-3-phosphoglyceric anhydride (or cyclic 1,3-phosphoglycerate) was never synthesized in vitro

Akhmadi and colleagues succeeded in synthesizing this compound and characterized the kinetic properties of PARK7’s action on this compound, revealing an outstanding catalytic efficiency of PARK7. They also demonstrate that cell extracts depend on PARK7 to prevent acylation of proteins caused by cyclic-3-phosphoglyceric anhydride.

To me this paper is very important, well-presented, and will undoubtedly be very extremely well received in the research community.

My main points of criticism concern the representation of prior work in abstract, results and the discussion. To avoid any impression of a hidden conflict of interest, I would like to indicate that I am the last author of the paper Heremans et al in PNAS2022.

The points that I would like to raise are the following

1. I have a problem with the way that the prior work of Heremans et al is presented. In Line 30, the authors state that 'It has been suggested that DJ-1 destroys cPGA,...'. Likewise, the authors state in line 61 and 63 that 'it was suggested' or 'proposed' that cPGA is the real substrate.

I do not think that this adequately represents the knowledge prior to the present paper, since Heremans and colleagues did not merely 'suggested' or 'proposed' this, but provided several lines of evidence that 1,3-bisphosphoglycerate converts into on cPGA (named c-1,3-PG in the paper) and that DJ-1 destroys this compound. These lines of evidence include estimations of its half-life that are close to what the authors observe, and the partial degradation into cyclic-2,3-phosphoglycerate (see figure 7 of Heremans et al). I think this should be clear in abstract and paper (in particular since it does not diminishes the value of the excellent paper under review)

For this reason, I would suggest the following changes

Line 30: 'Evidence has been provided that DJ-1 destroys cPGA, however this enzymatic activity has not been directly demonstrated'.

Lines 61 – 63: I think that it is again important to state that several lines of evidence were presented that 1,3-bisphosphoglycerate slowly converts into cPGA, and that PARK7/DJ1 acts on this compound.

2. Line 117: I do not think that Heremans claim that 'cyclic-2,3-phosphoglycerate is one of the major products of the reaction of 3PG with EDC'. They simply state that this compound is made (and the only compound that is clearly distinguishable from the 3PG by LC-MS). As can be seen in Fig. 7 of Heremans, the authors did not make any claim with regard to the relative proportion of hydrolysis of isomerization.

3. The discussion summarizes the findings of the paper very well, but it might make sense to mention that the measured half-life of cPGA (4min) is rather close to the half-life that was measured by Heremans et al for this compound when it is formed from 1,3-bisphosphoglycerate.

4. I would have liked to read the gene name PARK7 somewhere in the abstract.

Reviewer #3 (Remarks to the Author):

The authors of this study look into the chemistry and enzymatic regulation of a putative cyclic metabolite cPGA formed by presumably non-enzymatic intramolecular reaction by 1,3-bisphosphoglycerate, a metabolite that has been previously shown to modify amines in proteins and metabolites. Following a mechanistic rationalization of the in situ production of cPGA from chemical synthesis routes, the authors show that this electrophilic molecule can rapidly react with amine and thiols. More central to the manuscript is data suggestion that cysteine-containing enzyme DJ-1 is able to rapidly detoxify cPGA to 3PG, thus quenching its reactive acylation features. The enzymatic work is the main thrust of the work, however everything is performed with crude reaction products of 3PG with and electrophilic activating agent (EDC), and not cPGA alone; thus there are serious confounding factors to conclusions from these experiments as well as to the existence of cPGA itself in cells and the contribution to protein modification separate to or distinct of the parent metabolite 1,3-BPG. Oddly, the authors make many strong statements about the unlikely nature of 1,3-BPG directly acylating proteins, but do not do any experiments or comparisons themselves and ignore published examples of this reactivity in vitro, in cells and in animals. This general presentation of previous work through a skewed lens is prevalent the manuscript and is misleading at best. The authors then show a series of experiments where both the metabolite and enzyme are spiked into lysates to suggest that DJ-1 can regulate cPGA acylation of proteins using proteomics. As noted below, there are several technical issues with this study and the conclusions cannot be supported. While this is an interesting area and topic for study, the authors cannot show that cPGA even exists in cells, do not show that DJ-1 significantly regulates endogenous pgK and gK modifications (whether by 1,3-BPG or its putative product cPGA) and the primary evidence of reactivity of the proposed metabolite (which certainly is likely to exist) and its metabolism by DJ-1 are with crude mixtures of 3PG and an electrophilic activating group. Therefore, for these and more specific reasons below this manuscript would need major additional textual and experimental work to support the primary conclusions made by the authors.

Major Concerns:

1) The authors suggest that Heremans et al. discovered this acylation reaction between 1,3-BPG, and putatively cPGA, however they omit context and many previous observation relevant to the overarching goals of this study and numerous technical aspects presented in reference 14. Much of the conjecture on function and potential regulatory roles of this modification was described in that manuscript and should be reflected as such in the introduction and throughout the text.

For example, Line 164-165:, the authors here are implying that the reaction of cPGA, or its precursor 1,3-BPG – which are not differentiated in cellular experiments - with amines was demonstrated in ref 13. But this was already reported clearly in vitro and in cells (and in animals) in reference 14. The framing and attribution of mechanism is misaligned in much of the introduction of this manuscript.

2) The authors discuss that little or no pgK modification was observed on KLH following activation of 3PG with NHS/EDC, as measured by phosphate released from the adducted material with intestinal phosphatase. This method has not been previously demonstrated with this modification, to my knowledge, so its validity for which to draw such a conclusion is unclear. While this data is not shown, the logic that the reaction is forming cPGA in situ, which is then able to acylate target protein (KLH) would result in the same modified product (that is, pgK) as would directly modification by the EDC/NHS method. Therefore, it's not clear what the authors are discussing here. Since the 3PG-EDC/NHS method in reference 14 was used to successfully synthesize site-selective modified peptides, which were verified in chemical structure and purity by conventional methods, as well as acylated heterogeneous pgK proteins (BSA and KLH) for hapten generation, it is clear that either the analysis methods used by the authors (likely) or the specific preparations attempted by the authors were unsuccessful – not the approach. This should be clarified in the text or omitted entirely so as not to confuse readers or misrepresent that the previously published scheme is viable. The lead in to discussion the formation of cPGA is useful, but again, if that molecule were formed in situ it would likewise acylate the target protein and yield the same modification.

3) The authors, like Heremans, suggest that 'cPGA does not interfere with glycolysis' however this metabolite is derived from 1,3-BPG and therefore results in the destruction of this intermediate. The authors repeat this statement in a cyclic argument in the discussion: "Second, DJ-1 can prevent acylation of amines without destroying 1,3-BPG.¹³" In the authors model of Dj-1 action, the enzyme converts cPGA, which is derived from 1,3-BPG, into 3PG. How exactly is that mechanism 'preventing acylation of amines without destroying 1,3-BPG?' More importantly, this supposition is not evidence that 1,3-BPG is unlikely to acylate proteins, as is stated in this context.

4) The authors go on to make the statement: "Therefore 1,3-BPG is unlikely to acylate biological nucleophiles in direct reactions." However, they do no comparisons anywhere in the manuscript. By contrast, both ref. 14 and Chang & Moellering, *Analytical Chem*, 2016, show very clear direct acylation of small molecule nucleophiles, proteins and peptides with 1,3-BPG directly. This omission is glaring and misleading, and the authors should do a thorough job of comparing in this study and not make statements that are previously concluded unless they would like to provide experimental evidence.

5) I believe the key consideration for this manuscript is not whether cPGA can form, but rather whether there is any evidence that it is indeed forming in appreciable concentration within cells and whether it is the principal electrophile that causes formation of pgK modifications. Previous work by Moellering and Cravatt have shown that 1,3-BPG acts directly as an electrophile and can be quenched by nucleophiles (e.g., Chang & Moellering et al., *Analytical Chem*, 2016) without any chemical prodding to form cPGA. Here the authors don't really discuss the clear fact that 1,3-BPG itself is electrophilic, would react with the same nucleophiles to tracking in figure 3, and would show equivalent products on proteins or metabolites as cPGA. The text and figures should appropriately convey this or in cases where the authors believe they have convincingly demonstrated the differential reactivity and existence of 1,3-BPG from cPGA they should highlight these data and interpretations.

6) Throughout much of the biochemical characterizations in the paper, such as Figs 3 and 4, the authors make statements along the lines of “To gain insights into possible DJ-1 enzymatic activity towards cPGA, we incubated 0.5 mM of cPGA in phosphate buffer with different concentrations of DJ-1 for 1 minute and then 167 quantified the remaining cPGA by converting it into a thioester.” However, it is not clear what substrate they are using. The methods show that they are using a crude mixture of 3PG + EDC to produce cPGA in situ, in which case several species are being queried in parallel and many of the statements about cPGA having specific functions are indirect and unsupported. The authors should repeat this work with purified cPGA or discuss the many confounding factors in interpretation of any of these conclusions because crude mixtures are being used.

7) A key set of experiments are present in Figure 3, including 3C, from which the authors conclude that DJ-1 metabolizes cPGA to 3PG. However, the ¹H-NMR spectra of the product formed does not match the reference spectrum of 3PG. The authors should more convincingly demonstrate DJ-1-mediated conversion between cPGA to 3PG, and also show whether this conversion is possible with 1,3-BPG alone.

8) In the cellular studies of whether DJ-1 protects proteins from endogenous pgK and gK modifications, the authors used exogenous addition of cPGA. Again, this experiment could be performed with 1,3-BPG alone to compare and determine indeed if cPGA is a relevant acylating agent. More importantly, studies by both Heremens (ref 13) and Moellering and Cravatt (ref 14) demonstrated much higher detection of endogenous pgK and gK peptides via standard proteomic methods. If sensitivity is an issue, the authors should use the standard phosphoenrichment protocol reported in ref 14 to detect modified sites. Without detecting changes to endogenous pgK and gK sites, the authors cannot conclude that DJ-1 regulates acylation. The current conclusions that cPGA acts indiscriminantly contrasts to published profiles of endogenous sites that clearly argue for site-specific modification of protein lysines. The authors are oddly ignoring these data and instead reporting sub-par proteomics data and only informed by artificially spike in experimental set ups to generate their model. These studies need to be repeated to query endogenous pgK and gK modifications in cells.

9) In addition, there are no experiments in this paper that demonstrate that cPGA accumulates naturally to appreciable levels in cells. Without this demonstration all of the protective functions of DJ-1 and supposition that cPGA is the bona fide acylating reagent in cellular systems is conjecture.

We thank all the reviewers for their helpful comments. We revised the manuscript taking into account the reviewer's comments. We modified Figure 4 to include experiments demonstrating that DJ-1 can prevent acylation of GAPDH by 1,3-BPG in vitro without decreasing the concentration of 1,3-BPG. We also added a new Supplementary Figure 6 showing that DJ-1 completely converts cPGA into 3PG and a new Supplementary Figure 8 with information related to development and validation of anti-3PG antibodies.

Reviewer #1 (Remarks to the Author)

The authors report a kinetic and cellular study of DJ-1 activity against cyclic 1,3 phosphoglycerate anhydride (cPGA). This electrophile was previously proposed by Heremans et al (PMID: 35046029) to form spontaneously from 1,3 bisphosphoglycerate and to damage cellular nucleophiles. That same report showed that DJ-1 and its bacterial homolog YajL robustly defend proteins against this modification, which Heremans et al proposed was due to DJ-1's action on this proposed metabolite. In this report, the authors test this hypothesis by generating cPGA with a clever in vitro method that permits them to measure DJ-1 and YajL catalytic activity against this substrate. Confirming the prior proposal, they find that both proteins are highly efficient at converting cPGA to 3-phosphoglycerate, with k_{cat}/K_m values of 10⁶-10⁷. These catalytic efficiencies are several orders of magnitude higher than previously reported DJ-1 methylglyoxalase values and support the contention that this is a physiologically significant activity for DJ-1. They then corroborate Heremans et al.'s results showing that DJ-1 and YajL strongly reduce the amount of glycerate modification on diverse cellular proteins. In all, this is an impressive study that succeeds in generating an unstable, reactive metabolite in sufficient quantity to assay DJ-1 activity in vitro. The results confirm prior reports but also advance the field by permitting in vitro study of this activity, which is likely to be important for understanding DJ-1's function. The manuscript is clearly written and well-illustrated with a high degree of experimental rigor. My principal reservation is that the extent that this clearly demonstrated molecular activity is responsible for the much-studied cytoprotective activities of DJ-1 is still unknown, and there are some statements in the manuscript that should be softened in light of this. This and some other points for the authors to consider are detailed below.

1) The title of the manuscript is "DJ-1 protects cells by catalyzing the hydrolysis of cyclic 3-phosphoglyceric anhydride - a highly reactive glycolytic byproduct", but there are no data in the paper that address cellular protection by DJ-1. To be clear, this reviewer thinks it is likely that the activity measured by the authors makes an important contribution to DJ-1 biology and the authors present data establishing protection of proteins from cPGA modification, but this is different from cytoprotection. This is an essential distinction because the degree to which this new activity is responsible for established aspects of DJ-1 cytoprotection (e.g. anti-oxidative stress, modulation of the PTEN/Akt, ASK1, and other pathways) is not known and likely will be an important direction for future research. I think it would suffice for the authors to state that DJ-1 protects proteins from damage by cyclic 1,3 phosphoglycerate, which is directly addressed by their data.

We agree and have changed the title to "*DJ-1 protects proteins from acylation by catalyzing the hydrolysis of cyclic 3-phosphoglyceric anhydride - a highly reactive glycolytic byproduct*"

2) Related to (1), there are several places in the manuscript where the authors state that activity against cPGA is the "true molecular function of DJ-1". I find the data presented in this manuscript showing that

DJ-1 is highly active against cPGA to be quite convincing, which is exciting and opens a new chapter in the study of DJ-1. However, statements of “true molecular function” in the manuscript are sweeping and should be softened or removed. This also affects the final paragraph of the Results section (line 273, p.9). Like GAPDH (PMID: 20727968, 21895736), it seems likely that DJ-1 has multiple functions in cells, and multifunctionality (i.e. moonlighting) is a common theme in proteins (PMID: 29203708, 20144902). The GAPDH example is instructive: like DJ-1, it is an oligomeric cysteine-dependent protein that acts on triose phosphates and is subject to diverse post-translational modifications that alter its cellular function. The results presented by the authors are impressive enough that they do not require categorical statements of “true” function. This work provides a firm foundation upon which the role of this newly discovered DJ-1 activity can be further investigated.

DJ-1 has been ascribed an extremely large number of functions and physiological roles ranging from deglycase of DNA, amino acids and proteins to redox sensor, proteasome regulator and of course a participant of a large number of signaling pathways. In our opinion, it is impossible that DJ-1 performs so many physiologically relevant functions in cell, especially since none of the presumed signaling functions have a molecular/material background in form of enzymatic activity, well established binding partner or anything else. And while there are some proteins that have more than one function, these are exceptions, and in our opinion there is not enough evidence to suggest that DJ-1 is one of these proteins. We firmly believe that DJ-1 has one major function that should satisfy the following three criteria:

- 1) DJ-1 should have a molecular function that can be rigorously assessed/quantified (e.g. enzymatic activity or forming a complex with another protein).
- 2) This function should ideally explain an early onset of Parkinsonism in individuals lacking functional DJ-1.
- 3) Given the high degree of structural similarity between bacterial and human DJ-1, this function should be conserved in these organisms.

Before the discovery of its hydrolase function, none of the functions ascribed to DJ-1 satisfied all three of these criteria. This is why we use the expression “true molecular function” when discussing literature data. Nonetheless, we softened some sentences in the Discussion (lines 314-317) based on Reviewer 1’s suggestion: *“This activity is essential to prevent acylation of biomolecules by cPGA providing a potential mechanistic link to cytoprotective effects of DJ-1. This work provides a firm foundation upon which the role of cPGA hydrolase activity of DJ-1 in neuroprotection can be further investigated.”*

3) P. 2 “However, no signaling pathways or binding partners have been unequivocally established for DJ-1, leaving the mechanism of neuroprotection by DJ-1 an open question”. I acknowledge that “unequivocally” allows room for debate, but there are several high-quality reports that connect DJ-1 to the PTEN/Akt pathway (PMID: 15766664, many others), the ASK1 pathway (PMID: 19293155, 20385180), and several other pathways. It would be fair to rephrase to state that no clear, consensus molecular mechanism for DJ-1 cytoprotection currently exists.

This is related to (2) above. How is DJ-1 connected to PTEN/Akt or other pathways, that is, what is the underlying mechanism for this connection? In the case of PTEN, a redox-sensitive Tyr phosphatase with active site cysteine (like DJ-1) and a cluster of positively charged amino acids in active center recognizing phosphate (like DJ-1), it is possible that cPGA can inactivate PTEN in a direct reaction with its catalytic cysteine. If that is the case, DJ-1 might show an apparent synergy with this pathway by preventing PTEN

inactivation by cPGA. However, this “function” would be based on DJ-1’s true function as cPGA hydrolase. Moreover, there is another significant problem with ascribing signaling functions to DJ-1: a nearly identical protein is expressed by bacterial cells, which lack any of the mammalian signaling pathways.

Therefore, instead of citing several papers reporting DJ-1 involvement in various signaling pathways without underlying molecular mechanisms for modulating these pathways or their cytoprotective effects, we chose to state that “*no signaling pathways or binding partners have been unequivocally established for DJ-1, leaving the mechanism of neuroprotection by DJ-1 an open question*”. However, if Reviewer 1 insists that we cite these papers before concluding that “*no clear, consensus molecular mechanism for DJ-1 cytoprotection currently exists,*” we will do so in the final version of the manuscript.

4) There are some typographical errors (e.g. neutralized in Fig. 1, etc) that should be corrected.

Thank you. We have corrected Figure 1 and a few other typographical errors in the latest version of the manuscript.

Reviewer #2 (Remarks to the Author)

DJ1/PARK7 mutations are the cause of autosomal recessive forms of early-onset Parkinson's disease, but the molecular cause has remained elusive for a long time. In 2022, Heremans and colleagues found that this protein prevents acylation of metabolites and proteins by 1,3-bisphosphoglycerate. Evidence was presented that this acylation proceeds via cyclic-1,3-phosphoglycerate (referred to in the present work as cyclic-3-phosphoglyceric anhydride (cPGA). However, the enzymatic activity was never directly demonstrated and the cyclic-3-phosphoglyceric anhydride (or cyclic 1,3-phosphoglycerate) was never synthesized in vitro. Akhmadi and colleagues succeeded in synthesizing this compound and characterized the kinetic properties of PARK7's action on this compound, revealing an outstanding catalytic efficiency of PARK7. They also demonstrate that cell extracts depend on PARK7 to prevent acylation of proteins caused by cyclic-3-phosphoglyceric anhydride. To me this paper is very important, well-presented, and will undoubtedly be very extremely well received in the research community. My main points of criticism concern the representation of prior work in abstract, results and the discussion. To avoid any impression of a hidden conflict of interest, I would like to indicate that I am the last author of the paper Heremans et al in PNAS2022. The points that I would like to raise are the following

1. I have a problem with the way that the prior work of Heremans et al is presented. In Line 30, the authors state that 'It has been suggested that DJ-1 destroys cPGA,...'. Likewise, the authors state in line 61 and 63 that 'it was suggested' or 'proposed' that cPGA is the real substrate. I do not think that this adequately represents the knowledge prior to the present paper, since Heremans and colleagues did not merely 'suggested' or 'proposed' this, but provided several lines of evidence that 1,3-bisphosphoglycerate converts into on cPGA (named c-1,3-PG in the paper) and that DJ-1 destroys this compound. These lines of evidence include estimations of its half-life that are close to what the authors observe, and the partial degradation into cyclic-2,3-phosphoglycerate (see figure 7 of Heremans et al). I think this should be clear in abstract and paper (in particular since it does not diminishes the value of the excellent paper under review). For this reason, I would suggest the following changes Line 30: 'Evidence has been provided that DJ-1 destroys cPGA, however this enzymatic activity has not been directly demonstrated'. Lines 61 – 63: I think that it is again important to state that several lines of evidence were presented that 1,3-bisphosphoglycerate slowly converts into cPGA, and that PARK7/DJ1 acts on this compound.

We agree completely and introduced the following changes in abstract (line 31):

"Several lines of evidence pointed out that DJ-1 destroys cPGA, however this..." and introduction (line 62):
"This intermediate product has not been identified, but based on several kinetic experiments it was proposed that 1,3-BPG undergoes a slow reaction of cyclization..."

We added the following sentence at the end of the third paragraph (lines 65-67):

"Several lines of evidence indicate that DJ-1 and its homologs inactivate cPGA by hydrolysis explaining how DJ-1 prevents acylation of biomolecules by 1,3-BPG."

2. Line 117: I do not think that Heremans claim that 'cyclic-2,3-phosphoglycerate is one of the major products of the reaction of 3PG with EDC'. They simply state that this compound is made (and the only compound that is clearly distinguishable from the 3PG by LC-MS). As can be seen in Fig. 7 of Heremans, the authors did not make any claim with regard to the relative proportion of hydrolysis of isomerization.

We rephrased this sentence (lines 118-119) to reflect the fact that no claim regarding the proportion of 2,3-cyclic phosphodiester among all products has been made in PNAS2022 paper: *“Since Heremans et al.¹³ demonstrated that 2,3-cyclic phosphodiester is one of the products of reaction of 3PG with EDC we hypothesized that cPGA can form at the different pH levels used by Heremans et al. (pH 5.5) and in our study (pH ~2.5) but may subsequently decompose via different pH-dependent mechanisms.”*

3. The discussion summarizes the findings of the paper very well, but it might make sense to mention that the measured half-life of cPGA (4min) is rather close to the half-life that was measured by Heremans et al for this compound when it is formed from 1,3-bisphosphoglycerate.

We modified the discussion (lines 269-270) as follows: *“As expected, cPGA is highly unstable with a half-life of ~4 minutes which is similar to a half-life of ~8 minutes estimated by steady-state approximation of the kinetic profile of cPGA during continuous production of 1,3-BPG.”*

4. I would have liked to read the gene name PARK7 somewhere in the abstract.

It is common to refer to the gene as *PARK7*, and to its protein product as DJ-1, so we followed this convention in our paper. We now refer to *PARK7* in the first sentence of the abstract.

Reviewer #3 (Remarks to the Author):

The authors of this study look into the chemistry and enzymatic regulation of a putative cyclic metabolite cPGA formed by presumably non-enzymatic intramolecular reaction by 1,3-bisphosphoglycerate, a metabolite that has been previously shown to modify amines in proteins and metabolites. Following a mechanistic rationalization of the in situ production of cPGA from chemical synthesis routes, the authors show that this electrophilic molecule can rapidly react with amine and thiols. More central to the manuscript is data suggestion that cysteine-containing enzyme DJ-1 is able to rapidly detoxify cPGA to 3PG, thus quenching its reactive acylation features. The enzymatic work is the main thrust of the work, however everything is performed with crude reaction products of 3PG with and electrophilic activating agent (EDC), and not cPGA alone; thus there are serious confounding factors to conclusions from these experiments as well as to the existence of cPGA itself in cells and the contribution to protein modification separate to or distinct of the parent metabolite 1,3-BPG. Oddly, the authors make many strong statements about the unlikely nature of 1,3-BPG directly acylating proteins, but do not do any experiments or comparisons themselves and ignore published examples of this reactivity in vitro, in cells and in animals. This general presentation of previous work through a skewed lens is prevalent the manuscript and is misleading at best. The authors then show a series of experiments where both the metabolite and enzyme are spiked into lysates to suggest that DJ-1 can regulate cPGA acylation of proteins using proteomics. As noted below, there are several technical issues with this study and the conclusions cannot be supported. While this is an interesting area and topic for study, the authors cannot show that cPGA even exists in cells, do not show that DJ-1 significantly regulates endogenous pgK and gK modifications (whether by 1,3-BPG or its putative product cPGA) and the primary evidence of reactivity of the proposed metabolite (which certainly is likely to exist) and its metabolism by DJ-1 are with crude mixtures of 3PG and an electrophilic activating group. Therefore, for these and more specific reasons below this manuscript would need major additional textual and experimental work to support the primary conclusions made by the authors.

We feel that Reviewer 3 ignores nearly all the well-documented results from the Heremans et al. PNAS2022 paper, insisting that we repeat many of them. Moreover, they make several unfounded claims, repeatedly citing two papers that are less relevant to our study than those we cite. In some cases, these papers lack any evidence for the claims made by Reviewer 3. Therefore, we are not sure we can convince them with additional evidence or argument. Nonetheless, we attempted to address some of their criticisms by performing several additional experiments to support the main conclusions of our manuscript.

Major Concerns:

1) The authors suggest that Heremans et al. discovered this acylation reaction between 1,3-BPG, and putatively cPGA, however they omit context and many previous observation relevant to the overarching goals of this study and numerous technical aspects presented in reference 14. Much of the conjecture on function and potential regulatory roles of this modification was described in that manuscript and should be reflected as such in the introduction and throughout the text. For example, Line 164-165:, the authors here are implying that the reaction of cPGA, or its precursor 1,3-BPG – which are not differentiated in cellular experiments - with amines was demonstrated in ref 13. But this was already reported clearly in vitro and in cells (and in animals) in reference 14. The framing and attribution of mechanism is misaligned in much of the introduction of this manuscript.

The introduction aims solely at discussing DJ-1 and its potential functions, as this is the focus of this work. Therefore, we chose to cite a more relevant study as DJ-1 is not addressed in Ref. 14. Discussion of the regulatory/functional aspects of pgK modifications described in Ref. 14 would unnecessarily shift the focus of the manuscript. Neither we nor Heremans et al. studied the functional significance of these modifications. Moreover, rather than assuming that acylation of most abundant proteins and metabolites by glyceric and phosphoglyceric acid necessarily plays a regulatory role, we favor the more realistic hypothesis that presumes spontaneous formation of cPGA from 1,3-BPG, followed by an indiscriminate acylation of all cellular nucleophiles based on their relative abundance, solvent exposure and nucleophilicity.

Regarding Line 164-165, the claim that cPGA and 1,3-BPG are not differentiated in cellular experiments is incorrect. Cellular cPGA levels were manipulated in the PNAS2022 paper by deleting (Fig. 1-4, PNAS2022 paper) and re-expressing (Fig. 5 PNAS22 paper) DJ-1, an enzyme whose function is to destroy this reactive metabolite. Again, we cite a much more relevant work as there is nothing about DJ-1 or cPGA in Ref. 14.

2) The authors discuss that little or no pgK modification was observed on KLH following activation of 3PG with NHS/EDC, as measured by phosphate released from the adducted material with intestinal phosphatase. This method has not been previously demonstrated with this modification, to my knowledge, so its validity for which to draw such a conclusion is unclear. While this data is not shown, the logic that the reaction is forming cPGA in situ, which is then able to acylate target protein (KLH) would result in the same modified product (that is, pgK) as would directly modification by the EDC/NHS method. Therefore, it's not clear what the authors are discussing here. Since the 3PG-EDC/NHS method in reference 14 was used to successfully synthesize site-selective modified peptides, which were verified in chemical structure and purity by conventional methods, as well as acylated heterogeneous pgK proteins (BSA and KLH) for hapten generation, it is clear that either the analysis methods used by the authors (likely) or the specific preparations attempted by the authors were unsuccessful – not the approach. This should be clarified in the text or omitted entirely so as not to confuse readers or misrepresent that the previously published scheme is viable. The lead in to discussion the formation of cPGA is useful, but again, if that molecule were formed in situ it would likewise acylate the target protein and yield the same modification.

This point is not central to the main conclusions of our paper, but we would like to note the following. Given what we uncovered about the reaction of EDC with 3PG, it is clear that this reaction produces a strong acylating agent (cPGA) that decomposes very quickly on its own. Quick decomposition of cPGA is a significant problem since it results in products other than 3PG (such as 2,3-phosphodiester). Therefore, using the protocol from Ref. 14 that involves a 15 min pre-incubation of 3PG with a large excess of EDC/NHS at pH~5 is absolutely inadequate, since a large fraction of 3PG will be quickly converted to 2,3-phosphodiester of glyceric acid which will then react with EDC/NHS again. Once the pre-incubation is complete, the target protein will be modified not only with 3PG but also (or even predominantly) with various derivatives of cPGA decomposition. Therefore, the protocol in Ref. 14 will not “result in the same modification.”

There is ample evidence to support our conclusions. First, we could not detect 3PG adducts by Western blot using the protocol in Ref. 14 despite seeing a clear upward band shift (Supplementary Figure 8). Second, according to PNAS2022 paper, 2,3-phosphodiester of glyceric acid is resistant to hydrolysis by

alkaline phosphatase, which explains why we detected little or no phosphate release after phosphatase treatment of KLH modified with 3PG according to protocol from Ref. 14.

Once we better understood the reaction of EDC with 3PG, we acylated GAPDH by cPGA to achieve very high levels of modification (quantification of phosphate released by alkaline phosphatase indicated the presence of at least five 3PG molecules per molecule of GAPDH, indicating this method is sound). We used heavily acylated GAPDH to produce antibodies that specifically recognize 3PG modified proteins (Fig. 4a). Using these antibodies, we directly compared the acylation efficiency of our protocol with the procedure published in Ref. 14 and now include these results in new Supplementary Figure 8. This should alleviate any reader confusion as they now have data informing their choice of protocol.

As a side note, the protocol in Ref. 14 is likely to work for solid phase peptide synthesis but is poorly designed for protein modification by 3PG. It uses 1 mg/ml of KLH in PBS (~1.5 μ M) and states that KLH was modified with 20 equivalents of 3PG-NHS. Using 20 molar equivalents of 3PG-NHS (30 μ M) does not make sense, as this would modify only a small fraction of Lys residues on KLH (there are 300-600 Lys residues on the surface of KLH molecule). Probably, some other equivalence unit (mass? volume?) is implied that the reader must figure out on their own. In any case, much higher concentrations of 3PG-NHS are necessary, but they will acidify the sample (PBS has low buffer capacity) leading to lower acylation efficiency.

Finally, if peptides are modified using the procedure in Ref. 14 and end up being labelled with 2,3-cyclic phosphodiester instead of 3PG, they are likely to be transformed into a mixture of 2PD- and 3PD-modified peptides during deprotection and cleavage from the resin. However, the same cannot be expected for modified proteins which are always at near-neutral pH. On the other hand, if protein modifications are assayed by LC-MS/MS, 2,3-cyclic phosphodiester modifications may still hydrolyze to result in 2PG and 3PG modifications during the relatively harsh and lengthy sample preparation procedures. Therefore, LC-MS/MS analysis may show 3PG modifications even if the protein is actually modified by 2,3-cyclic phosphodiester.

3) The authors, like Heremens, suggest that ‘cPGA does not interfere with glycolysis’ however this metabolite is derived from 1,3-BPG and therefore results in the destruction of this intermediate. The authors repeat this statement in a cyclic argument in the discussion: “Second, DJ-1 can prevent acylation of amines without destroying 1,3-BPG.^{13”} In the authors model of Dj-1 action, the enzyme converts cPGA, which is derived from 1,3-BPG, into 3PG. How exactly is that mechanism ‘preventing acylation of amines without destroying 1,3-BPG?’ More importantly, this supposition is not evidence that 1,3-BPG is unlikely to acylate proteins, as is stated in this context.

We never stated that ‘cPGA does not interfere with glycolysis.’ Reviewer 3 likely refers to a sentence in the Introduction (lines 60-61) essentially asserting that DJ-1 does not interfere with glycolysis. The phrase “DJ-1 can prevent acylation of amines without destroying 1,3-BPG” simply means that the presence of DJ-1 does not cause any additional destruction of 1,3-BPG but prevents the formation of acylation products. This is because the conversion of 1,3-BPG to cPGA is an irreversible spontaneous reaction that will occur at the same rate, regardless of the presence or absence of DJ-1. However, since DJ-1 destroys cPGA but not 1,3-BPG, the mechanism of prevention of acylation is clear: we show that DJ-1 has an extremely high hydrolase activity that destroys cPGA before it can react with cellular nucleophiles. Conversely, if almost all the acylation by 1,3-BPG is prevented by DJ-1, it is clear that 1,3-BPG does not acylate proteins directly

but does so indirectly via the formation of cPGA. Since other reviewers did not find these parts confusing or contradictory, we would like to leave this text as is.

Let us assume that 1,3-BPG decomposes at a certain rate producing cPGA. If cPGA is a predominant acylating agent in this system, the addition of DJ-1 will:

1. Prevent most of the acylation by destroying cPGA
2. Will not decrease concentration of 1,3-BPG

Both outcomes were reported by Heremans et al. and now by us (Fig. 4b and c). If 1,3-BPG acylates molecules in a single step bimolecular reaction as Reviewer 3 suggests, how can the presence or absence of small amounts of DJ-1 influence acylation by 1,3-BPG? The only way would be for DJ-1 to destroy 1,3-BPG or hydrolyze adducts of 3PG, but our data and that of Heremans et al. demonstrate that DJ-1 does not destroy 1,3-BPG. In fact, our data suggests that the presence of DJ-1 leads to higher steady state levels of 1,3-BPG at long incubation times (Fig. 4b). If 1,3-BPG acylates biomolecules directly, the presence of DJ-1 would not change acylation levels (or should increase them because of higher steady-state levels of 1,3-BPG), but all available evidence is inconsistent with this. The only way to implicate 1,3-BPG in direct acylation would be to ignore the dramatic increases in acylation levels reported in the PNAS2022 paper for DJ-1 knockout animals and cells and our data that show an exceptionally high activity of DJ-1 as cPGA hydrolase. We do not think this provides an adequate ground for a productive discussion.

4) The authors go on to make the statement: “Therefore 1,3-BPG is unlikely to acylate biological nucleophiles in direct reactions.” However, they do no comparisons anywhere in the manuscript. By contrast, both ref. 14 and Chang & Moellering, Analytical Chem, 2016, show very clear direct acylation of small molecule nucleophiles, proteins and peptides with 1,3-BPG directly. This omission is glaring and misleading, and the authors should do a thorough job of comparing in this study and not make statements that are previously concluded unless they would like to provide experimental evidence.

We would like to point out that if any two molecules react with each other, *it does not mean that the reaction is direct (i.e. it is a single-step bimolecular reaction) and does not involve a rate-determining step (RDS) of reactive intermediate formation.*

Of note, several lines of evidence implicating the mechanism with RDS of reactive intermediate formation in the process of acylation by 1,3-BPG are presented in PNAS2022 paper (Fig. 6G and Fig. 7, PNAS2022 paper). In contrast, both ref. 14 and Chang & Moellering, Analytical Chem, 2016, assume a direct acylation because nothing was known about cPGA formation from 1,3-BPG at that time. Neither of these references shows “very clear direct acylation.” To prove that the acylation by 1,3-BPG is direct, it would be necessary to show (at the minimum) a linear dependence of the rate of 1,3-BPG consumption on the concentration of nucleophiles, but this was not done in either paper. Ideally, second order kinetic constants should have been determined for reactions of 1,3-BPG with nucleophiles to gauge the reactivity (as we did for cPGA in this study), but again this was not done in either paper. It is possible that in the presence of extremely high concentrations of nucleophiles (e.g. ~2 M hydroxylamine as in Analytical Chem 2016 paper), direct acylation by 1,3-BPG will become prevalent, but this needs to be shown in an independent study via carefully planned kinetic experiments designed to detect and quantify the contribution of acylation by cPGA.

The study of Heremans et al. demonstrates that 1,3-BPG is relatively inert towards nucleophiles. For example, the presence of 1-5 mM of cysteamine does not change the rate of its spontaneous decay (Fig. 6F, PNAS22 paper) but prevents acylation (Fig. 6G, PNAS2022 paper). These data indicate that acylation is not direct but involves a reactive intermediate. More importantly, the addition of sub-stoichiometric amounts of DJ-1 protein to a system that produces 1,3-BPG continuously completely prevents acylation of small molecules (Fig. 6C and Fig. 7H, PNAS2022 paper) and proteins (Fig. 6D, PNAS22 paper). We generated 1,3-BPG continuously to achieve a high steady-state concentrations of 5-6 mM (Fig. 4b) and were able to detect acylation of GAPDH by Western blot. However, this acylation was completely prevented by sub-stoichiometric amounts of DJ-1 (Fig. 4c, this manuscript) in agreement with the conclusions of the PNAS 2022 paper. Based on these combined data, *it is impossible to implicate 1,3-BPG in direct acylation*. The same conclusion was made in PNAS2022 paper: “*This reveals a so far unknown chemical property of 1,3-BPG: it does not directly modify amino groups efficiently. Rather, this modification proceeds mostly via a reactive intermediate, which can be eliminated by PARK7 before it reacts with amino groups of metabolites and proteins*”.

In view of the above, we assert that we have, in fact, made no glaring omissions and nothing in our manuscript is misleading. We are frustrated at having spent precious time and money repeating well-designed experiments from the PNAS 2022 paper to come to exactly the same conclusions.

Due to the high concentrations of nucleophiles in the cell, 1,3-BPG can, in principle, participate in direct reactions of acylation in a cellular context. The extent to which this may occur is unclear at the moment, however based on the available data involving deletion of DJ-1 (Figs 1-4 of PNAS22 paper) and DJ-1 re-expression in knockout cells (Fig 5 of PNAS22 paper) it is clear that most if not all of the acylation is indirect and occurs via cPGA formation. In the absence of any sufficient evidence implicating 1,3-BPG in direct acylation and in face of overwhelming evidence implicating cPGA, we will leave this part of the text as is.

5) I believe the key consideration for this manuscript is not whether cPGA can form, but rather whether there is any evidence that it is indeed forming in appreciable concentration within cells and whether it is the principal electrophile that causes formation of pgK modifications. Previous work by Moellering and Cravatt have shown that 1,3-BPG acts directly as an electrophile and can be quenched by nucleophiles (e.g., Chang & Moellering et al., Analytical Chem, 2016) without any chemical prodding to form cPGA. Here the authors don't really discuss the clear fact that 1,3-BPG itself is electrophilic, would react with the same nucleophiles to tracking in figure 3, and would show equivalent products on proteins or metabolites as cPGA. The text and figures should appropriately convey this or in cases where the authors believe they have convincingly demonstrated the differential reactivity and existence of 1,3-BPG from cPGA they should highlight these data and interpretations.

This point has been sufficient addressed in 4) above. Previous work (e.g., Chang & Moellering et al., Analytical Chem, 2016) assumed direct acylation by 1,3-BPG since nothing was known about cPGA formation at the time, but 1,3-BPG reactivity has not been characterized in terms of reaction rates, reaction orders or kinetic constants. Therefore, this work does not implicate 1,3-BPG as a bona fide acylation agent, especially since nucleophiles quench acylation without affecting 1,3-BPG levels (PNAS2022 paper). Granted, the fact that acylation by 1,3-BPG is not direct is unexpected, however it does not mean that all the evidence suggesting the indirect nature of acylation should be simply ignored. We again point out how well our study agrees with the PNAS2022 paper: the substrate for DJ-1 (cPGA) was proposed *because 1,3-BPG could not be implicated in direct acylation*. We were able to synthesize this

molecule and showed that the suppositions made in the PNAS2022 paper are true. A consideration of these facts together leads to little doubt about the true enzymatic function of DJ-1.

With regards to 1,3-BPG reactivity, we do not discuss it because based on our results and those in the PNAS2022 paper, most (if not all) acylation by 1,3-BPG is mediated by formation of cPGA both in vitro and in vivo.

6) Throughout much of the biochemical characterizations in the paper, such as Figs 3 and 4, the authors make statements along the lines of “To gain insights into possible DJ-1 enzymatic activity towards cPGA, we incubated 0.5 mM of cPGA in phosphate buffer with different concentrations of DJ-1 for 1 minute and then quantified the remaining cPGA by converting it into a thioester.” However, it is not clear what substrate they are using. The methods show that they are using a crude mixture of 3PG + EDC to produce cPGA in situ, in which case several species are being queried in parallel and many of the statements about cPGA having specific functions are indirect and unsupported. The authors should repeat this work with purified cPGA or discuss the many confounding factors in interpretation of any of these conclusions because crude mixtures are being used.

The reviewer’s assertion is incorrect. We clearly show that equimolar mixture of 3PG, EDC and HCl immediately produces cPGA with near 100% yield with no trace of EDC left. Based on NMR spectra recorded immediately after mixing of reagents, there are no traces of any other molecules present (Supplementary Figure 1). A complete absence of EDC in stoichiometric mixtures of EDC and 3PG is warranted since we know that cPGA will start decomposing immediately, and products of its decomposition will react with any remaining EDC, effectively eliminating any leftover traces. This is especially clear from Supplementary Figure 2 that shows that the presence of a few percent of unreacted EDC would be clearly visible in NMR spectra, but we see no leftover EDC in equimolar mixture of 3-PG and EDC.

However, even assuming that traces of EDC are present, how would these affect our results? EDC is orders of magnitude less reactive at the neutral pH at which enzymes are assayed and most of it would remain unreacted. For example, assuming that ~5% of EDC remains unreacted reason, this would correspond to 25 μ M if we use 0.5 mM cPGA. This EDC will remain mostly unreacted with only a small percentage reacting with the NAC that is present in ~100 times excess. Still, we decided to test this idea by assaying DJ-1 with 0.5 mM of cPGA plus 10% and 20% of extra added EDC. As expected (Fig. below), small changes in kinetics are evident with a 20% EDC excess, because it reacts slowly with 3PG and NAC, leading to additional thioester formation. Kinetics in the presence of DJ-1 remains roughly the same with or without excess EDC present. Clearly, any potential problems can be avoided by ensuring that no excess of EDC is used when preparing cPGA from 3PG.

As Reviewer 3 must understand, purifying cPGA is not possible due to its short-lived nature (3.9 min half-time).

7) A key set of experiments are present in Figure 3, including 3C, from which the authors conclude that DJ-1 metabolizes cPGA to 3PG. However, the ¹H-NMR spectra of the product formed does not match the reference spectrum of 3PG. The authors should more convincingly demonstrate DJ-1-mediated conversion between cPGA to 3PG, and also show whether this conversion is possible with 1,3-BPG alone.

Fig. 3C aims to show the existence of cPGA at neutral pH and that the addition of DJ-1 causes its disappearance. Since cPGA decomposes quickly giving us only a few minutes to record the NMR spectrum, we had to use ~50 mM cPGA, which likely immediately inactivated most of DJ-1 so that decomposition products other than 3PG are also present. However, analyzing only reaction products is an easier task since NMR spectra can be collected for longer times so lower concentrations of cPGA (e.g. 6 mM) can be used in experiments. We added a new Supplementary Figure 6 showing a complete conversion of cPGA into 3PG catalyzed by DJ-1. Conversion of 1,3-BPG to 3PG via the destruction of a reactive intermediate by DJ-1 has already been reported in the PNAS2022 paper

8) In the cellular studies of whether DJ-1 protects proteins from endogenous pgK and gK modifications, the authors used exogenous addition of cPGA. Again, this experiment could be performed with 1,3-BPG alone to compare and determine indeed if cPGA is a relevant acylating agent. More importantly, studies by both Heremans (ref 13) and Moellering and Cravatt (ref 14) demonstrated much higher detection of endogenous pgK and gK peptides via standard proteomic methods. If sensitivity is an issue, the authors should use the standard phosphoenrichment protocol reported in ref 14 to detect modified sites. Without detecting changes to endogenous pgK and gK sites, the authors cannot conclude that DJ-1 regulates acylation. The current conclusions that cPGA acts indiscriminantly contrasts to published profiles of endogenous sites that clearly argue for site-specific modification of protein lysines. The authors are oddly ignoring these data and instead reporting sub-par proteomics data and only informed by artificially spike in experimental set ups to generate their model. These studies need to be repeated to query endogenous pgK and gK modifications in cells.

We see little value in repeating the experiments of Heremans et al. who demonstrated extremely strong increases in protein and metabolite acylation upon knockout of DJ-1 in three different organisms. Given this data, we did not set out to determine whether DJ-1 regulates endogenous acylation levels. If Reviewer 3 doubts the data and conclusions from PNAS2022, it is incumbent on them to explain why these experiments should be repeated.

The true nature of cPGA could not be established in the PNAS2022 paper with 100% certainty, but we had a unique possibility to test the main hypothesis of Heremans et al. directly using synthetic cPGA, a true substrate of DJ-1. We did not ask whether DJ-1 prevents acylation of lysines (abundantly clear from the PNAS2022 paper), but how it achieves this. Our data clearly indicate that DJ-1 is both necessary and sufficient to inactivate cPGA and protect proteins from acylation in cytosolic extracts which has proteins and DJ-1 present in the same ratio as inside the cell. If DJ-1 prevents acylation of proteins in cytosolic extract caused by large concentrations of cPGA, it will protect proteins to the same extent or better within a living cell that constantly produces cPGA as a side product. Using 1,3-BPG in a similar experiment would only repeat many of the experiments reported in PNAS2022 paper.

As discussed above, there is abundant evidence that most if not all acylation by 1,3-BPG is cPGA-mediated and indiscriminate. Our views regarding indiscriminate nature of acylation are based on the reactivity pattern of cPGA established in this study (it is clear, for example, that amino acids will effectively compete with proteins for cPGA) and on data from the PNAS2022 paper that demonstrates that acylation correlates well with protein abundance (Fig. 4E, PNAS2022) and that individual proteins can be acylated on most of the exposed lysines or N-termini (Fig. 4B-D, PNAS2022 paper). Therefore, acylation by cPGA will be discriminate only inasmuch as surface lysines differ in abundance, and (to a lesser extent) solvent accessibility and nucleophilicity. This was also one of the conclusions of PNAS2022 paper.

9) In addition, there are no experiments in this paper that demonstrate that cPGA accumulates naturally to appreciable levels in cells. Without this demonstration all of the protective functions of DJ-1 and supposition that cPGA is the bona fide acylating reagent in cellular systems is conjecture.

This is a spurious assertion at best. cPGA will not accumulate to appreciable levels in cells due to its extremely high intrinsic reactivity and due to catalysis by DJ-1. There are numerous examples of reactive metabolites that cannot be detected directly (hydroxyl, peroxy, and thiyl radicals, many metabolites of nitric oxide such as peroxyxynitrite) and are assessed exclusively by indirect methods in thousands of papers every year. We (Fig. 4c) and Heremans et al. demonstrated that production of cPGA is an *intrinsic property* of 1,3-BPG. Saying that cPGA does not exist in cells is akin to saying 1,3-BPG does not exist in cells, an obviously false claim. The many-fold increase in 3PG adducts in DJ-1 deficient cells from three different organisms reported previously is best explained rationally by the exceptionally high cPGA hydrolase activity of DJ-1 characterized in this study. Any other hypothesis stretches plausibility considering the evidence presented in our manuscript.

REVIEWERS' COMMENTS

Reviewer #1 (Remarks to the Author):

The authors have largely addressed my concerns in their revised manuscript. I appreciate that they have moderated some of the statements about the function of DJ-1, leaving room for additional future work to determine to what extent DJ-1's cPGA hydrolase activity accounts for its established roles in stress response and various signalling pathways. The authors' ask in their rebuttal whether I am requesting that they cite various literature references that I provided in my first review. I am not requesting these citations-I included them to provide concrete examples of my suggestions. I am more favorably inclined towards the view that DJ-1 is a multifunctional protein than the authors are, but of course this diversity of opinions is what helps science move forward.

Reviewer #2 (Remarks to the Author):

The authors have responded adequately to my points.

Reviewer #3 (Remarks to the Author):

In their revised manuscript and response, the authors have thoroughly considered and provided responses to comments and concerns raised. Overall, I believe these comments alleviate significant experimental concerns. This paper is improved and should be published with final edits to the text to shore up claims being made and the context in which these claims are presented. The latter is important and was the focus of comments from both reviewers: many statements in the original submission were either unclear to the reader (several for this reviewer), or there were statements and claims that were or are over- or misstated. This may not have been clear from my original comments and critiques, but many of the responses provided by the authors in their response were more detailed and than the original text. I provide specific comments and suggestions to improve the final text for clarity such that this work can be adequately considered by readers in the proper context of work in the field.

Major Comments:

1) In their revised manuscript and response, the authors have gone to additional lengths to study the modification of protein antigens to produce anti-pgK antibodies. The comparison of the 3pg/nhs/edc protocol, and suggestion that it could produce a mixture of adducts on antigen proteins is an important addition and should be clearly stated as such for the readers.

The original and current text states: “However, we saw little to no modification of KLH by 3PG as assessed by phosphate release from purified KLH by calf intestinal phosphatase.” This statement, and others surrounding it, indicate that these proteins are not modified by 3PG, when instead you are suggesting that they are likely modified by a mixture of 3PG and 2,3-phosphodiester. It is relevant and important to clarify, especially in the context of the procedure developed by the authors in this work.

2) Throughout the authors should consider using ‘pgK’ to refer to and annotate these sites, especially in figures. This was the nomenclature used in previous reports on these modifications and is customary when referring to posttranslational modifications (such as AcK, Ksucc, pT/S/Y and so on). The authors should consider adopting the same in figures and text in lieu of the long form N-3-phosphoglyceroyl lysine.

3) The discussions around what are the probable and plausible acylating molecules that ultimately lead to pgK modifications from glucose-derived 1,3-BPG should be revised to adequately weigh the evidence at hand. The authors, following the work in the 2022 PNAS report by Heremans, have provided strong evidence that cPGA is formed at some rate from 1,3-BPG, that this more reactive intermediate can acylate proteins and that DJ-1 regulates this process. However, the discussion surrounding the contribution of these two metabolites in cells cannot be fully clarified (eg, cPGA cannot be differentiated in cellular context at all) and therefore statements in absolute terms go too far, such as the following in their response “it is impossible to implicate 1,3-BPG in direct acylation,” and in the discussion of the current text:

“Finally, the many-fold increase in acylation 255 levels of proteins and metabolites in cells and organisms lacking DJ-113 cannot be clearly explained by assuming 256 that 1,3-BPG is a bona fide acylation agent in vivo. Therefore, 1,3-BPG is unlikely to acylate biological 257 nucleophiles in direct reactions. The slow spontaneous formation of highly reactive cPGA from 1,3-BPG 258 (Figure 1a) suggested by Heremans et al.¹³ reconciles these and other prior findings, but most importantly, it 259 introduces cPGA as a probable, unrecognized metabolic byproduct of glycolysis with a high potential to 260 indiscriminately acylate cellular nucleophiles.”

I do not believe making an absolute statement that 1,3-BPG cannot participate in direct protein acylation is supported here. The most relevant consideration is merely one of analogous chemistry that has been established:

The authors suggest that the reactive cPGA can acylate proteins, whereas the acylphosphate precursor 1,3-BPG cannot. An ideal test for this would be a molecule that contains the reactive moiety in 1,3-BPG but that cannot rearrange to the more active cyclic anhydride. Such a molecule exists, and has been characterized – this molecule is acetylphosphate, which is involved in prokaryotic metabolism and shown through several studies to directly acylate protein amines in vitro, in cells and in vivo. Most notably the

work by Weinert and Choudhary (PMID 23830618 and subsequent studies). Combined with the established reactivity of amines toward this functionality, these data refute the notion that a precursor metabolite like 1,3-BPG cannot acylate proteins. Acetylphosphate directly modifies recombinant proteins and levels of AcK modifications change in response to modulation of metabolite levels in cells and organisms. Thus, there is ample reason to believe that 1,3-BPG, while distinct, can carry out the same chemistry, and making statements to the contrary in absolute terms is not warranted.

Moreover, since cPGA is derived from 1,3-BPG it is the level of these two reactive intermediates that regulate acylation of biomolecules (assuming a steady state rate of conversion from the latter to the former). The absolute statements suggesting that 1,3-BPG is not the relevant molecule to consider in these modifications could be therefore confusing and misleading, as it is the level of 1,3-BPG that regulates acylation.

The authors should include/revise a discussion to this effect and temper their absolute statements in light of all evidence at hand. I don't think these changes lessen the impact of the work at hand, but it will avoid making statements that cannot be fully supported.

4) The use of 1,3-BPG modified GAPDH as an antigen is interesting and looks to be effective. It is established that GAPDH from recombinant sources does carry endogenous pgK modifications. This likely explains cross-reactivity of Abs with the unmodified GAPDH. It does raise questions about the efficacy of counter-purifying antibodies using unmodified GAPDH, and also this method likely produces a somewhat narrow pool of pAbs that are biased toward GAPDH sites. The authors should include these caveats in their discussion of this and related methods.

5) On the topic of where and how specific publications in the field are cited, it is appropriate to cite the initial work describing 1,3-BPG as a reactive molecule that leads to the formation of pgK modifications in the context of the results by Heremans and connection with the proposed role for DJ-1. It is odd to instead only first mention that work only in the context of a somewhat peripheral methodological comment. That was the focus of my original comment on citation and attribution.

Reviewer #1 (Remarks to the Author):

The authors have largely addressed my concerns in their revised manuscript. I appreciate that they have moderated some of the statements about the function of DJ-1, leaving room for additional future work to determine to what extent DJ-1's cPGA hydrolase activity accounts for its established roles in stress response and various signalling pathways. The authors' ask in their rebuttal whether I am requesting that they cite various literature references that I provided in my first review. I am not requesting these citations-I included them to provide concrete examples of my suggestions. I am more favorably inclined towards the view that DJ-1 is a multifunctional protein than the authors are, but of course this diversity of opinions is what helps science move forward.

We thank Reviewer 1 for comments and efforts to improve the quality of our work

Reviewer #2 (Remarks to the Author): The authors have responded adequately to my points.

We thank Reviewer 2 for comments and efforts to improve the quality of our work

Reviewer #3 (Remarks to the Author): In their revised manuscript and response, the authors have thoroughly considered and provided responses to comments and concerns raised. Overall, I believe these comments alleviate significant experimental concerns. This paper is improved and should be published with final edits to the text to shore up claims being made and the context in which these claims are presented. The latter is important and was the focus of comments from both reviewers: many statements in the original submission were either unclear to the reader (several for this reviewer), or there were statements and claims that were or are over- or misstated. This may not have been clear from my original comments and critiques, but many of the responses provided by the authors in their response were more detailed and than the original text. I provide specific comments and suggestions to improve the final text for clarity such that this work can be adequately considered by readers in the proper context of work in the field.

We thank Reviewer 3 for comments and efforts to improve the quality of our work

Major Comments:

1) In their revised manuscript and response, the authors have gone to additional lengths to study the modification of protein antigens to produce anti-pgK antibodies. The comparison of the 3pg/nhs/edc protocol, and suggestion that it could produce a mixture of adducts on antigen proteins is an important addition and should be clearly stated as such for the readers. The original and current text states: "However, we saw little to no modification of KLH by 3PG as assessed by phosphate release

from purified KLH by calf intestinal phosphatase.” This statement, and others surrounding it, indicate that these proteins are not modified by 3PG, when instead you are suggesting that they are likely modified by a mixture of 3PG and 2,3-phosphodiester. It is relevant and important to clarify, especially in the context of the procedure developed by the authors in this work.

We modified the original sentence as follows: *“However, based on low amount of phosphate released from modified KLH by calf intestinal phosphatase, we found this protocol to be inefficient”* to reflect our observations more accurately. We introduced the following text (lines 201-205) to outline the observed differences in acylation outcomes: *“We acylated GAPDH with cPGA and used it to obtain rabbit polyclonal antibodies specific for pgK modifications (Supplementary Figure 8 and Fig. 4a). In line with our initial observations, standard EDC/NHS protocol which involves 15 min pre-incubation of 3PG with EDC/NHS¹⁴ was found to be less effective and likely resulted in modification of GAPDH by a mixture of 3PG and 2,3-phosphodiester of glyceric acid (Supplementary Figure 8).”*

2) Throughout the authors should consider using ‘pgK’ to refer to and annotate these sites, especially in figures. This was the nomenclature used in previous reports on these modifications and is customary when referring to posttranslational modifications (such as AcK, Ksucc, pT/S/Y and so on). The authors should consider adopting the same in figures and text in lieu of the long form N-3-phosphoglyceroyl lysine.

We adopted the use of pgK to abbreviate 3-phosphoglyceroyl lysine.

3) The discussions around what are the probable and plausible acylating molecules that ultimately lead to pgK modifications from glucose-derived 1,3-BPG should be revised to adequately weigh the evidence at hand. The authors, following the work in the 2022 PNAS report by Heremans, have provided strong evidence that cPGA is formed at some rate from 1,3-BPG, that this more reactive intermediate can acylate proteins and that DJ-1 regulates this process. However, the discussion surrounding the contribution of these two metabolites in cells cannot be fully clarified (eg, cPGA cannot be differentiated in cellular context at all) and therefore statements in absolute terms go too far, such as the following in their response “it is impossible to implicate 1,3-BPG in direct acylation,” and in the discussion of the current text “Finally, the many-fold increase in acylation levels of proteins and metabolites in cells and organisms lacking DJ-113 cannot be clearly explained by assuming that 1,3-BPG is a bona fide acylation agent in vivo. Therefore, 1,3-BPG is unlikely to acylate biological nucleophiles in direct reactions. The slow spontaneous formation of highly reactive cPGA from 1,3-BPG (Figure 1a) suggested by Heremans et al.¹³ reconciles these and other prior findings, but most importantly, it introduces cPGA as a probable, unrecognized metabolic byproduct of glycolysis with a high potential to 260 indiscriminately acylate cellular nucleophiles.” I do not believe making an absolute statement that 1,3-BPG cannot participate in direct protein acylation is supported here. The most relevant consideration is merely one of analogous chemistry that has been established: The authors suggest that the reactive cPGA can acylate proteins, whereas the acylphosphate precursor 1,3-BPG cannot. An ideal test for this would be a molecule that contains the reactive moiety in 1,3-BPG but that cannot rearrange to the more active cyclic anhydride. Such a molecule exists, and has been characterized – this molecule is acetylphosphate, which is involved in prokaryotic metabolism and shown through several studies to directly acylate protein amines in vitro, in cells and in vivo. Most notably the work by Weinert and Choudhary (PMID 23830618 and subsequent studies). Combined with the established reactivity of amines toward this functionality, these data refute the notion that a precursor metabolite like 1,3-BPG cannot acylate proteins. Acetylphosphate directly modifies recombinant proteins and levels of AcK modifications change in response to modulation of metabolite levels in cells and organisms. Thus, there is ample reason to believe that 1,3-BPG, while

distinct, can carry out the same chemistry, and making statements to the contrary in absolute terms is not warranted. Moreover, since cPGA is derived from 1,3-BPG it is the level of these two reactive intermediates that regulate acylation of biomolecules (assuming a steady state rate of conversion from the latter to the former). The absolute statements suggesting that 1,3-BPG is not the relevant molecule to consider in these modifications could be therefore confusing and misleading, as it is the level of 1,3-BPG that regulates acylation. The authors should include/revise a discussion to this effect and temper their absolute statements in light of all evidence at hand. I don't think these changes lessen the impact of the work at hand, but it will avoid making statements that cannot be fully supported.

While the published data suggest that most of the acylation inside the cell is due to cPGA formation, we agree that making absolute statements regarding acylation properties of 1,3-BPG is not warranted. We removed the following sentence from the Discussion: “*Therefore, 1,3-BPG is unlikely to acylate biological nucleophiles in direct reactions*”. We modified the first paragraph in the Discussion to reflect more clearly our views and the evidence that supports them and to remove any unsubstantiated claims.

4) The use of 1,3-BPG modified GAPDH as an antigen is interesting and looks to be effective. It is established that GAPDH from recombinant sources does carry endogenous pgK modifications. This likely explains cross-reactivity of Abs with the unmodified GAPDH. It does raise questions about the efficacy of counter-purifying antibodies using unmodified GAPDH, and also this method likely produces a somewhat narrow pool of pAbs that are biased toward GAPDH sites. The authors should include these caveats in their discussion of this and related methods.

Testing of rabbit serum (Figure 8, *left panel*) suggested that our antibodies recognized unmodified GAPDH. This could be either due to the presence of endogenous pgK modifications and/or due to the fact that a subset of antibodies recognize epitopes of GAPDH that cannot be acylated (e.g. due to lack of surface lysines) and therefore are identical in modified and unmodified GAPDH. The latter seemed more probable to us, so we used immobilized unmodified GAPDH for additional purification. We have (most likely) lost a tiny fraction of antibodies that bound to the GAPDH with endogenous pgK modifications, but we definitely were able to remove a subset of antibodies that recognizes unmodified GAPDH since these counter-purified antibodies nearly lost their ability to produce a signal from the unmodified GAPDH (and the remaining weak signal, when observed, may indeed come from endogenous pgK modifications). This can be seen on all blots where counter-purified antibodies were used (Fig. 4a, Fig. 4c, and Supplementary Figure 8, *right panel*). This purification step was not accompanied by a notable loss in the amount of antibodies, but after this additional step antibodies still produced a strong signal only if proteins were treated with enough cPGA. Thus, this procedure deemed effective. A couple of other approaches produced less satisfactory results.

After the treatment of cytosolic extracts with cPGA, antibodies recognize a very large number of modified proteins which is evident from more or less uniform distribution of a signal in treated samples (Fig. 4e). There is no single prominent band around 35 kDa to suggest that antibodies are strongly biased towards modified GAPDH. Of course, these antibodies may have limitations, but they are of sufficient quality to detect acylation of GAPDH by 1,3-BPG and are quite efficient in detecting differences in acylation of many proteins in knockout and wild type cells (Fig. 4c). Therefore, we request to exclude any discussion of efficiency/limitations of these antibodies: while we cannot guarantee that these antibodies will recognize pgK modification in every possible situation, this is not really necessary for the type of experiments reported in this manuscript.

5) On the topic of where and how specific publications in the field are cited, it is appropriate to cite the initial work describing 1,3-BPG as a reactive molecule that leads to the formation of pgK

modifications in the context of the results by Heremens and connection with the proposed role for DJ-1. It is odd to instead only first mention that work only in the context of a somewhat peripheral methodological comment. That was the focus of my original comment on citation and attribution.

We modified the Introduction as follows (lines 60-63): *“1,3-BPG has been known to acylate lysine side chains in proteins resulting in 3-phosphoglyceroyl-lysine (pgK) modifications.¹⁴ However, since DJ-1 neither uses 1,3-BPG as a common substrate, nor repairs pgK residues in modified proteins,¹³ it was suggested that the true substrate of DJ-1 is a previously unknown reactive intermediate product of 1,3-BPG decomposition.”*